# Self-supervision improves Multi-teacher Distillation

## Abstract

Knowledge Distillation (KD) has evolved from compressing large models to enhancing the performance of models with the same capacity. Multi-teacher distillation extends this paradigm by amalgamating knowledge from multiple expert models into a single student. Multi-teacher knowledge distillation aims to create a powerful student model by amalgamating knowledge from multiple expert teachers. However, existing frameworks constrain the student to learn exclusively from the teachers' representations, overlooking valuable supervisory signals inherent in the data itself. In this work, we introduce **Self-supervised Feature Aggregation (SeFA)**, a novel paradigm that addresses this limitation by synergistically combining multi-teacher distillation with self-supervised learning. SeFA formulates the training as a multi-task learning problem, optimizing the student's representations for both alignment with its teachers and performance on a data-driven, self-supervised task. We conduct extensive evaluations across a diverse set of tasks, including image classification, transfer learning, domain adaptation, image retrieval, and dense prediction. SeFA consistently outperforms state-of-the-art baselines, achieving average improvements of 6.11% on classification, 8.87% on image retrieval, and 6.44% on dense prediction tasks. Beyond these empirical gains, our comprehensive analysis demonstrates SeFA's robustness across various teacher combinations and architectures, establishing a more effective paradigm for knowledge distillation.

## 1 Introduction

Since its introduction in computer vision, knowledge distillation (KD) has been adapted for a wide range of applications. Early KD methods focused primarily on compressing the knowledge of a large teacher model into a smaller student model for task-specific scenarios. Born-Again Networks (BAN) Furlanello et al. (2018) introduced a new perspective by demonstrating that distillation between models of equal size can also yield performance gains for vision models.

Traditional KD approaches, including BAN, typically rely on KL divergence or cross-entropy–based loss functions, where the student learns to match the teacher's output distribution. While effective for task-specific distillation, these methods are less suited for learning a general purpose student. More recent works address this by distilling from multiple teachers into a single student, often using distance- or similarity-based losses to align feature representations. The goal in this setting is for the student to inherit desirable priors from all teachers. These approaches fall under the category of multi-teacher distillation. However, early multi-teacher distillation methods lacked mechanisms to prevent dominance by a single teacher. UNIC (the existing state-of-the-art) Sariyildiz et al. (2024) addressed this by introducing teacher-dropping regularization, where the student randomly zeroes out the loss term associated with its closest teacher in normalized feature space. While this promotes diversity among the learned representations, it does not guarantee that the resulting features are more generalizable than those being suppressed. As a result, the student may still acquire non-beneficial representations. This a key limitation of existing multi-teacher distillation approaches in computer vision. In this work, we argue that **self-supervised approaches, which capture task-agnostic and non-trivial features, can serve as a complementary signal to multi-teacher distillation, guiding the student toward more beneficial and generalizable representations**.

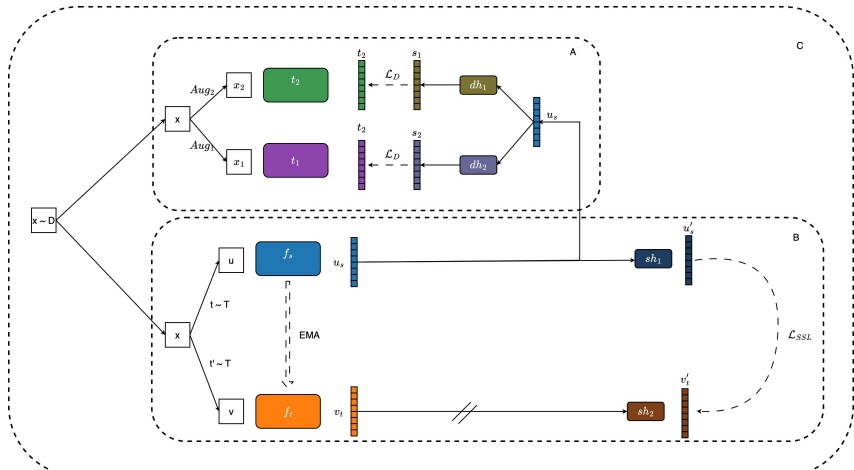

Figure 1: Overview of SeFA within the distillation framework using two teachers (Box A), where DINO Caron et al. (2021) is used for self-supervised learning (Box B). $t_1$ and $t_2$ denote the two teachers, with $Aug_1$ and $Aug_2$ representing the teacher-specific augmentations applied to $t_1$ and $t_2$, respectively. The student encoder is denoted by $f_s$, and $f_t$ refers to its exponential moving average (EMA) copy, used exclusively for the self-supervised objective. Here, $L_D$ represents the distillation loss $L_{\text{Distillation}}$ as defined in Eq. 5. $dh_1$ and $dh_2$ denote the two distillation heads that map the student encoder's representation to the corresponding teacher's representation space. Meanwhile, $sh_1$ and $sh_2$ represent the self-supervised heads used for the self-supervised objective. We further expand on this figure in Appendix A. Post-training we utilize features from $f_s$.

The improvement of multi-teacher distillation is of great interest especially because of the rise of many pre-trained vision models. We conduct a comprehensive analysis of the improved multi-teacher distillation process from multiple teachers, evaluating our models on various vision tasks, including image-level classification, video segmentation, transfer learning, image retrieval, robustness and adversarial classifications. Furthermore, we ablate student model size, number and size of teachers, self-supervised loss used in conjunction with the multi-teacher distillation loss. To the best our knowledge we are the first to provide a comprehensive study on how to pick teachers for multi-teacher distillation for improved performance.

We begin in Section 3 by discussing the shortcomings of existing state-of-the-art vision based multi-teacher distillation methods and presenting a theoretical perspective on their limitations. Section 5 then presents our empirical evaluation, where we compare the proposed approach against existing state-of-the-art multi-teacher distillation baselines across coarse- and fine-grained classification, image retrieval, and dense prediction tasks such as video instance segmentation. In Section 5.2, we further characterize the behavior of our method under varying conditions using five different general-purpose teachers, three distinct frameworks, and two student architectural configurations.

## 2 RELATED WORKS

In Buciluǎ et al. (2006) the authors showed that the knowledge from multiple models or an ensemble model could be effectively transferred into a model of same or smaller capacity. In a typical distilation setup, both the student and the teacher operate in the same domain and the student attempts to match teacher's representation. In the beginning, this approach was used to distill larger models into smaller models, but Xie et al. (2020) suggested using student and teacher architectures of the same capacity. Simultaneously, knowledge distillation eventually grew from one student trying to match one teacher to multiple teachers. In early multi-teacher distillation methods, the common strategy was to match the average predictions of multiple teachers. However, this approach limits the ability to exploit the complementary nature of different teachers' representations. Zuchniak (2023) provides an overview of multi-teacher distillation, and proposes that instead of matching an ensemble of teachers, the student can match the features of each individual teacher via some learned non-shared mapping from the representation space of the student to each teacher. This has become of the basis of more

recent works, such as Theia Shang et al. (2024), UNIC: Universal Classification Models Sariyildiz et al. (2024), AM-RADIO: Agglomerative Vision Foundation Model Reduce All Domains Into One Ranzinger et al. (2024), SAM-CLIP Wang et al. (2024), and Open-Vocabulary Segment Anything Model (SAM) Yuan et al. (2024), aim to combine the semantics captured by Contrastive language Image Pretraining (CLIP) Radford et al. (2021) with the localization capabilities of models like DINOv2 Oquab et al. (2023) or SAM Kirillov et al. (2023).

Multi-teacher distillation assumes that when vision models are trained long enough on a substantial amount of data they learn diverse and meaningful representations. Also, these representations can be distilled from these models into a singular model of similar or lower capacity. This knowledge transfer is achieved by minimizing a feature-similarity loss between the student and teachers. Common objectives include Mean Squared Error (MSE), L1 loss, and Cosine Similarity. For example, Open-Vocabulary SAM Yuan et al. (2024) adopts an MSE loss and SAM-CLIP Wang et al. (2024) employs cosine similarity, while state-of-the-art methods like Theia Shang et al. (2024), AM-RADIO Ranzinger et al. (2024), and UNIC Sariyildiz et al. (2024) use a compound loss that combines smooth L1 and cosine similarity.

Since multi-teacher distillation is based on the assumption that different teachers have different representations, hence alignment with one would mean misalignment with other teacher(s). Furthermore, distilling all features without filtering or regularization could lead to features from one teacher dominating and leading to a poor performing student method. UNIC Sariyildiz et al. (2024) empirically studies this phenomenon and was the first multi-teacher distillation approach to introduce a regularizer to mitigate dominance of one teacher's features over others. They introduced teacher-dropping regularization, wherein the most correlated teacher is randomly dropped to encourage the student to learn a more diverse set of representations. We further discuss this in the Section 3.

## 3 METHOD

### 3.1 MOTIVATION

As mentioend in Section 1, SAM-CLIP Wang et al. (2024), Open-vocabulary SAM Yuan et al. (2024), AM-RADIO Ranzinger et al. (2024), Theia Shang et al. (2024) only use similarity based losses such as smooth-L1, cosine similarity or MSE for multi-teacher distillation. Current state-of-the-art approaches such as UNIC Sariyildiz et al. (2024) typically rely on the traditional multi-teacher distillation frameworks along with a regularizer. For instance, UNIC (Universal Classification) employs a "teacher-dropping" regularizer, which encourages the student to learn from the least correlated teacher under the assumption that this exposes it to novel and complementary knowledge. However, without an explicit mechanism to evaluate which teacher representations are truly generalizable downstream, this strategy risks promoting diversity at the expense of utility—potentially steering the student toward a representation space that is diverse but suboptimal.

In UNIC Sariyildiz et al. (2024), for a given batch of data and a set of $T$ teachers, UNIC begins by computing a loss for each individual teacher $i \in \{1, \ldots, T\}$. This similarity loss, $\mathcal{L}_{\text{sim}}$, is formulated as a weighted combination of cosine similarity and smooth L1 loss:

$$\mathcal{L}_{\text{sim}}(\mathbf{x}; i) = \alpha \cdot \mathcal{L}_{\text{CosineSim}}(\mathbf{t}_i(x), h_{t_i}(\mathbf{z})) + (1 - \alpha) \cdot \mathcal{L}_{\text{SmoothL1}}(\mathbf{t}_i(x), h_{t_i}(\mathbf{z})), \quad (1)$$

where $h_{t_i}(\mathbf{z})$ is the student's prediction and $\mathbf{t}_i(x)$ is the normalized output from teacher $t_i$. As noted in prior work (Heo et al., 2019), normalizing teacher outputs is crucial for ensuring a balanced contribution from the ensemble. Next, UNIC generates a vector of teacher coefficients, $\mathbf{c} = [c_1, \ldots, c_T]$, by first identifying the teacher with the maximum loss on a given batch, $t^* = \arg\max_{t \in \{1, \ldots, T\}} \mathcal{L}_{\text{sim}}(\mathbf{x}; t)$. The coefficients $c_t$ are then determined as shown in Equation equation 2:

$$c_t = \begin{cases} 1 & \text{if } t = t^* \\ d_t & \text{if } t \neq t^* \end{cases} \quad \text{where } d_t \sim \text{Bernoulli}(1 - p_{\text{drop}}). \quad (2)$$

This strategy guarantees that the teacher providing the strongest learning signal for a sample is always included. The final loss, $\mathcal{L}_{\text{UNIC}}$, is the weighted sum of the per-teacher losses from Equation equation 1:

$$\mathcal{L}_{\text{UNIC}} = \sum_{i=1}^{T} c_i \cdot \mathcal{L}_{\text{sim}}(\mathbf{x}; i), \quad (3)$$

where the coefficients $c_i$ are treated as detached constants and do not contribute to the gradient calculation.

The design of the UNIC loss function presents two fundamental limitations that hinder student learning. First, it assumes teachers provide representations of uniform quality. In a realistic scenario with one informative and one noisy teacher, a student aligning with the former would be heavily penalized by its large loss against the latter. This can cause teacher-dropping mechanisms to discard the wrong teacher, compelling the student to learn a corrupted representation. Second, the student's learning is entirely constrained to the teachers' representations, precluding it from capturing valuable information directly from the input data. To remedy these shortcomings, we introduce Self-supervised Feature Aggregation (SeFA).

With SeFA the loss function to be as follows for a given input batch $x$:

$$\mathcal{L}_{\text{total}}(x) = \lambda_1 \mathcal{L}_{\text{SSL}}(x) + (1 - \lambda_1)\mathcal{L}_{\text{Distillation}}(x) \tag{4}$$

$$\mathcal{L}_{\text{Distillation}}(x) = \sum_{i=1}^{T} \mathcal{L}_{\text{sim}}(x; i) \tag{5}$$

$$\mathcal{L}_{\text{sim}}(\mathbf{x}; i) = [\alpha \times \mathcal{L}_{\text{CosineSim}}(\mathbf{t}_i(x), h_{t_i}(\mathbf{z}))] + [(1 - \alpha) \times \mathcal{L}_{\text{SmoothL1}}(\mathbf{t}_i(x), h_{t_i}(\mathbf{z}))] \tag{6}$$

In this reformulated loss, the student model is essentially solving a multi-task optimization problem (Eq. 4). It must find a set of features that are effective for solving the self-supervised task $L_{\text{SSL}}$ as well as structurally similar to the features of a strong pre-trained teacher $L_{\text{Distillation}}$.

Existing multi-teacher distillation methods compel a student model to learn exclusively from teacher representations. This approach overlooks a rich source of supervisory signal: the data itself. We argue that by failing to mine this data-specific information, student models are unnecessarily constrained and may fail to learn powerful, generalizable features.

To address this, we introduce a self-supervised learning (SSL) objective, which reframes the student's task as a multi-task optimization problem. The student must learn a feature representation that is simultaneously: (1) structurally similar to the teacher's features via a distillation loss, $L_{\text{Distillation}}$, and (2) effective for a pretext self-supervised task, via $L_{\text{SSL}}$.

SSL is uniquely suited for this role. Its objective functions are designed to learn fine-grained, task-agnostic semantic representations directly from the data. Moreover, mechanisms within modern SSL techniques, such as contrastive or reconstruction objectives, inherently prevent representational collapse to trivial solutions. Our central hypothesis is that this synergistic, multi-task approach allows the student to learn robust representations that are both semantically rich and aligned with the knowledge distilled from the teachers.

**Our Contribution:** In this paper, we argue that multi-teacher distillation approaches can benefit from multi-tasking with self-supervised objectives. To this end, we propose Self-supervised Feature Aggregation (SeFA), as introduced in the previous section. Similar to prior work in the field Sariyildiz et al. (2024), we begin by training SeFA with two teachers in Section 5. We observe that it consistently outperforms the existing state-of-the-art method UNIC across a variety of tasks. Beyond the conventional scenario of distilling a smaller student from a larger teacher, SeFA enables multi-teacher distillation in which the student can surpass its teachers on a wide range of computer vision tasks. Additionally, in Section 5.2, we establish a framework for selecting optimal teacher combinations in multi-teacher distillation.

## 4 EXPERIMENTAL SETUP

We start with a two-teacher multi-distillation setup. In each case we compare performance of the trained student against best performing teacher as well as the existing state-of-the art multi-teacher distilation method with teacher dropping regularization – UNIC Sariyildiz et al. (2024). With SeFA,

| Model | ImageNet Acc. |
|---|---|
| *Teacher Models* | |
| DINO ViT-S/16 | 76.09 |
| AugReg ViT-S/16 | **76.50** |
| SeFA DINO (*ours*) | 77.92 |
| SeFA UDI (*ours*) | 78.30 |
| SeFA iBOT (*ours*) | 76.90 |

Table 1: Top-1 accuracy on the ImageNet-1K validation set for ViT-S/16 models trained with various self-supervision techniques combined with multi-teacher distillation.

we argue that self-supervision alongside traditional multi-teacher distillation enhances regularization in multi-teacher distillation. We start by incorporating a self-supervised objective alongside the distillation loss. For the self-supervised component, we begin with DINO Caron et al. (2021) due to its state-of-the-art performance and fine-grained control over feature clustering via temperature scaling and the number of centroids. We ablate the choice of self-supervised regularizer in Section 5.2.

For the two teacher models we pick one self-supervised and a supervised model similar to Sariyildiz et al. (2024). For the self-supervised model we pick a ViT-S/16 Dosovitskiy et al. (2020) trained on ImageNet-1K using DINO Caron et al. (2021) for 800 epochs, and the second is a ViT-S/16 Dosovitskiy et al. (2020) trained on ImageNet-21K Ridnik et al. (2021) and ImageNet-1K Deng et al. (2009) (a total of 15M images) using state-of-the-art fully supervised learning with augmentation regularization (AugReg) Steiner et al. (2021). We then trained a ViT-S/16 using the above two teachers with UNIC Sariyildiz et al. (2024) as well as with **Se**lf-supervisied **F**eature **A**ggregation (denoted as SeFA) using the protocol mentioned in Section 4.

Models trained with self-supervised contrastive losses Caron et al. (2021); Zhou et al. (2022a); Su & Ji (2024) tend to emphasize low-frequency signals and produce features that enable linear separability based on semantic content Park et al. (2023). In contrast, large scale supervised models such as Steiner et al. (2021); Touvron et al. (2022) tend to rely more heavily on high-frequency image details. We pretrain a ViT S/16 using DINO and Augreg teachers on the ImageNet-1k dataset. To isolate the effect of better features from larger models or from other datasets during multi-teacher distillation, we train student and teacher models of same size and only using teachers trained on ImageNet datasets i.e ImageNet-1k or models pretrained on the larger ImageNet-21k Ridnik et al. (2021) followed by finetuning on ImageNet-1k. We further ablate the choice of the teacher models in Section 5.2.

Once pretrained, we train a linear layer on top of the frozen pre-trained student and use the performance obtained via linear probing from the downstream tasks (described in section X) to evaluate the effectiveness of our models and the baselines.This evaluates how linearly separable classes are in the learned feature space: strong performance by a linear classifier trained on frozen features suggests useful, transferable representations. Following prior work Caron et al. (2021); Chen* et al. (2021); Oquab et al. (2023); Shang et al. (2024); Sariyildiz et al. (2024), we evaluate our pre-trained encoders on ImageNet-1K validation set using a linear classifier. In addition we also study properties of the resulting features for retrieval, object discovery and transfer-learning.

To further validate the above results and rule out potential overfitting to the ImageNet-1K dataset Deng et al. (2009), we evaluate our pre-trained model on additional datasets, each justifying a unique property.ImageNet-A Hendrycks et al. (2021b) measures robustness to natural adversarial examples—real-world images that cause frequent misclassifications despite being semantically correct—revealing vulnerabilities in decision boundaries. ImageNet-C Hendrycks & Dietterich (2019) tests corruption robustness by applying common perturbations (e.g., noise, blur, weather effects), showing how performance degrades under distribution shift. ImageNet-R Hendrycks et al. (2021a) assesses out-of-distribution (OOD) robustness using artistic renditions (e.g., cartoons, paintings, sketches), exposing reliance on photographic priors. ImageNetV2 Recht et al. (2019), collected a decade later but with identical classes, indicates whether the encoder has learned durable semantic concepts rather than overfitting to dataset-specific visual statistics.

**Implementation details**    For both UNIC and SeFA, we pretrain models on the ImageNet dataset Deng et al. (2009) without labels. For fair comparison we use a batch size of 512 for 200 epochs, distributed over 4 GPUs using ViT-S/16 Dosovitskiy et al. (2020). For each experiment we perform random hyperparamter over five sets of hyperparameter for pre-training. To ensure a fair comparison, encoder gradients are frozen and a linear classifier is trained on the output features of the frozen network with a learning rate sweep over 45 values in $[10, 10^{-4}]$. We pre-train using five different hyperparameter configurations and then train a linear classifier for each, spanning 45 learning rates resulting in a total of 225 comabinations. The best-performing model for each method and pre-training epoch is selected from these 225 combinations, with results presented in Section 5.

| **Model** | IN1k-val | IN-V2 | IN-R | IN-A | IN-C ($\downarrow$) | Transfer |
|---|---|---|---|---|---|---|
| *Teacher Models* | | | | | | |
| DINO ViT-S/16 | 76.09 | 71.95 | 34.50 | 11.0 | 67.82 | 85.63 |
| AugReg. ViT-S/16 | 76.50 | 71.77 | 35.73 | 16.83 | 68.69 | 86.85 |
| Best Teacher | 76.50 | 71.95 | 35.73 | **16.83** | 67.82 | 86.85 |
| SeFA ViT-S/16 (*ours*) | **77.92** | **73.87** | **37.71** | 11.57 | **64.66** | **87.23** |
| UNIC ViT-S/16 | 71.17 | 67.07 | 27.71 | 8.23 | 71.30 | 84.06 |

Table 2: Top-1 accuracy on ImageNet and related datasets. Results are reported as percentages for all datasets, except ImageNet-C where we use the normalized mean corruption error (mCE) Hendrycks & Dietterich (2019). Higher values indicate better performance for accuracy, while lower values are better for mCE.

| **Pretrain** | $\mathcal{R}_{Oxford}$ | | $\mathcal{R}_{Paris}$ | |
|---|---|---|---|---|
| | M | H | M | H |
| DINO | 34.62 | 12.98 | 60.81 | 32.32 |
| Augreg | 28.94 | 9.71 | 63.41 | 38.24 |
| Best Teacher | 34.62 | 12.98 | 63.41 | **38.24** |
| SeFA | **36.57** | **14.43** | **63.5** | 35.6 |
| UNIC | 28.14 | 7.83 | 53.44 | 25.2 |

Table 3: Comparison of pretraining methods on ROx and RPar under Medium (M) and Hard (H) protocols using ViT-S/16 Dosovitskiy et al. (2020).

| Model | $\mathcal{J}_{\mathrm{m}}$ | $\mathcal{F}_{\mathrm{m}}$ | $(\mathcal{J}\&\mathcal{F})_{\mathrm{m}}$ |
|---|---|---|---|
| *Teacher Models* | | | |
| DINO ViT-S/16 | 59.94 | 63.31 | 61.62 |
| Augreg ViT-S/16 | 47.71 | 49.38 | 48.54 |
| Best teacher | 59.94 | 63.31 | 61.62 |
| SeFA ViT-S/16 (ours) | **60.36** | **64.20** | **62.28** |
| UNIC ViT-S/16 | 53.82 | 57.78 | 55.92 |

Table 4: Mean region similarity $\mathcal{J}_{\mathrm{m}}$, contour accuracy $\mathcal{F}_{\mathrm{m}}$, and combined $(\mathcal{J}\&\mathcal{F})_{\mathrm{m}}$ on DAVIS 2017. using ViT-S/16 Dosovitskiy et al. (2020).

## 5 RESULTS

### 5.1 DINO SELF-SUPERVISION AS REGULARIZER

Following the setup in Section 4, we train a ViT-S/16 student on ImageNet-1k and compare our proposed method, SeFA, against the state-of-the-art UNIC Sariyildiz et al. (2024) baseline. To evaluate the learned representations, we use a linear probing protocol on the frozen features across a comprehensive suite of benchmarks: the ImageNet-1k validation set, its variants (ImageNetV2, -R, -A), and four transfer learning datasets. For robustness, we also report the mean Corruption Error (mCE, lower is better) on ImageNet-C.

The results demonstrate that SeFA consistently outperforms the UNIC baseline across all ImageNet-based classification tasks. More notably, SeFA surpasses the performance of both of its teacher models on all benchmarks except ImageNet-A—a feat not achieved by UNIC. This highlights SeFA's ability not only to aggregate but also to refine and improve upon the knowledge distilled from its teachers. To further analyze results on the ImageNet-A dataset, instead of linear probing we perform

end-to-end finetuning for 100 epochs and observe the ImageNet-A accuracy increases to 19.85% with an ImageNet1k-val accuracy of 79.96%. We present more results on end-to-end finetuning and the protocol used in Appendix C.8.Consistent with prior work Caron et al. (2021); Chen* et al. (2021); Zhou et al. (2022a); Oquab et al. (2023); Sariyildiz et al. (2024), in addition to standard ImageNet classifications we also benchmark on image retrieval, object discovery and transfer-learning.

**Transfer Learning.** To assess the transferability of learned representations on ImageNet to other major datasets, we follow the evaluation protocol described in Appendix C. Specifically, we freeze the pretrained encoders and train a linear classifier on top of their features across four fine-grained image classification datasets: CIFAR-10, CIFAR-100 Krizhevsky et al. (2009), Flowers-102 Nilsback & Zisserman (2008), and IIIT-Pets Parkhi et al. (2012). After hyperparameter search, we present the best accuracy of the linear classifier in Table 2. From here, we observe that similar to the classification performance on ImageNet-1k and other ImageNet like datasets, our model improves performs better than both teachers involved by 0.40%. Furthermore, it improves by 3.17% over existing state-of-art for multi-teacher distillation i.e UNIC Sariyildiz et al. (2024).

**Video Segmentation.** We evaluate the frozen patch-level outputs of all the pre-trained models mentioend in Section 2 using the DAVIS-2017 video instance segmentation benchmark Pont-Tuset et al. (2017). Following prior work Caron et al. (2021); Zhou et al. (2022a); Su & Ji (2024), segmentation across video frames is performed using a nearest-neighbor clustering between patch embeddings of consecutive frames. This evaluation is conducted without any additional fine-tuning. We present the results for Video Object detection in Table 4. Despite not optimized for dense prediction tasks, our model achieves competitive results, suggesting that the learned representations retain spatial structure. We observe that SeFA overall improves by video segmentation over best performing teacher by 0.66% and by 6.36% over over existing state-of-art.

**Image Retrieval.** In Table 3, we present nearest-neighbor retrieval results using frozen pre-trained backbones on the revisited Oxford-5k and Paris-6k datasets Radenović et al. (2018). We observe that none of the teachers consistently performs best across both splits of the Oxford and Paris revisited datasets. In contrast, our SeFA-trained ViT-S/16 achieves state-of-the-art performance on three out of four metrics. Specifically, on the Oxford dataset, the SeFA ViT-S/16 improves by approximately 1.95 and 1.45 mAP on the medium and hard splits, respectively. Furtermore, SeFA improves over existing state-of-art UNIC by approximately 7% and 10% on the Oxford and Paris revisited datasets Radenović et al. (2018) respectively.

| Model | IN1k-val | IN V2 | IN-R | IN-A | IN-C ($\downarrow$) | Transfer |
|---|---|---|---|---|---|---|
| *Teacher Models* | | | | | | |
| DINO ViT-B/16 [†] | 77.99 | 74.13 | 38.49 | 15.33 | 61.27 | 88.70 |
| AugReg ViT-B/16 | 79.70 | 75.03 | 41.67 | 23.83 | 60.03 | 89.34 |
| Best Teacher | 79.70 | 75.03 | 41.67 | **23.83** | 60.03 | 89.34 |
| SeFA ViT-B/16 (*ours*) | **80.71** | **76.47** | **42.07** | 17.69 | **58.21** | **89.88** |
| UNIC ViT-B/16 | 76.32 | 72.03 | 36.77 | 13.69 | 64.51 | 84.79 |

Table 5: Similar to Table 2, we report Top-1 accuracy on ImageNet and related datasets using ViT B-16 Dosovitskiy et al. (2020). Results are given in percentage, except for ImageNet-C where we use the normalized mean corruption error (mCE) Hendrycks & Dietterich (2019). Higher accuracy values indicate better performance, while lower values are preferable for mCE.

## 5.2 ABLATION STUDY

In Section 5, we have already established that multi-teacher distillation with self-supervised regularization is a better approach for agglomerating multiple teachers into a single encoder for a variety vision tasks such as transfer learning, classification, image retrieval and segmentation. In this section, we probe further into SeFA to systematically understand nuances of using Self-supervision as regularization with Multi-teacher Distillation.

| Pretrain | $\mathcal{R}_{Oxford}$ | | $\mathcal{R}_{Paris}$ | |
|---|---|---|---|---|
| | M | H | M | H |
| DINO | 34.34 | 10.76 | 60.55 | 32.98 |
| Augreg | 28.29 | 9.67 | 60.13 | 35.56 |
| Best Teacher | 34.34 | 10.76 | 60.55 | 35.56 |
| SeFA | **36.38** | **13.42** | **64.72** | **37.36** |
| UNIC | 35.04 | 10.81 | 58.78 | 29.36 |

Table 6: Image retrieval results using ViT-B/16 with DINO self-supervision and multi-teacher distillation.

| Model | $\mathcal{J}_m$ | $\mathcal{F}_m$ | $(\mathcal{J}\&\mathcal{F})_m$ |
|---|---|---|---|
| *Teacher Models* | | | |
| DINO | 60.6 | 63.9 | 62.3 |
| Augreg | 50.8 | 54.01 | 54.01 |
| Best teacher | 60.6 | 63.9 | 62.3 |
| SeFA (ours) | **60.7** | **64.01** | **62.4** |
| UNIC | 50.90 | 55.31 | 53.10 |

Table 7: Similar to Table 4, we provide metrics on DAVIS 2017 using ViT B/16 using DINO self-supervision with multi-teacher distillation.

**Change in self-supervised framework for regularization** To investigate whether other self-supervised approaches can regularize an agglomerated encoder during multi-teacher distilation, we keep the teachers and multi-teacher distillation loss constant while ablating the self-supervised approach used for regularization in SeFA. In addition to DINO Caron et al. (2021), we evaluate iBoT Zhou et al. (2022a) and UDI Su & Ji (2024). iBoT Zhou et al. (2022a) combines masked image modeling with self-distillation objective of DINO Caron et al. (2021), while UDI Su & Ji (2024) enriches representations through context-aligned semantic constraints via self-attention and an additional class token ($cls^+$) to produce multi-modal predictions in addition to the same DINO Caron et al. (2021) objective. We present the results in Table 1. The results indicate that coupling multi-teacher distillation with a self-supervised regularization loss improves performance over each individual teacher used for distillation. We train DINO- and UDI-based SeFA for 200 epochs each, and iBoT-based SeFA for 800 epochs. iBoT requires longer training to first optimize its contrastive objective before effectively learning masked image modeling. Furthermore, DINO is the most compute efficient to train followed by iBOT and UDI Su & Ji (2024).

| Teacher Combination | Pre-training + *Linear* | | CKA | Similarity | In1k Acc.(%) |
|---|---|---|---|---|---|
| | 100 PTE | 200 PTE | | | |
| **High Similarity (CKA $\approx$ 0.8)** | | | | | |
| DINO + iBOT | 76.57 | 76.75 | 0.8190 | **High** | 77.05 |
| DINO + MUGS | 75.80 | 75.93 | 0.8051 | **High** | 76.09 |
| iBOT + MUGS | 76.56 | 76.72 | 0.8117 | **High** | 77.05 |
| **Moderate Similarity (CKA $\approx$ 0.5)** | | | | | |
| DINO + AugReg | 77.80 | 77.92 | 0.5082 | **Moderate** | 76.50 |
| iBOT + AugReg | 78.12 | 78.34 | 0.5243 | **Moderate** | 77.05 |
| MUGS + AugReg | 77.14 | 77.64 | 0.5066 | **Moderate** | 76.50 |
| **Low Similarity (CKA $\approx$ 0.2)** | | | | | |
| DINO + DeiT-III | 78.87 | 79.33 | 0.2001 | **Low** | 79.80 |
| iBOT + DeiT-III | 78.96 | 79.41 | 0.2054 | **Low** | 79.80 |
| AugReg + DeiT-III | 78.33 | 78.87 | 0.1989 | **Low** | 79.80 |

Table 8: Top-1 Accuracy on the ImageNet-1K validation dataset using SeFA ViT-S/16 models distilled from teacher pairs. Cells are green when student accuracy exceeds the best teacher, and red otherwise. CKA values quantify representational similarity, with regimes defined as **Low** (CKA $\approx$ 0.2), **Moderate** ( CKA $\approx$ 0.5), and **High** (CKA $\approx$ 0.8).

**Changing teacher combinations** To study the impact of teacher diversity, we examine teachers through the lens of representation similarity using linear Centered Kernel Alignment (CKA) Kornblith et al. (2019) and using five diverse ViT-S/16 or equivalent foundation models - DINO Caron et al.

(2021), iBOT Zhou et al. (2022a), **MU**lti-**G**ranular **S**elfsupervised learning or MUGS Zhou et al. (2022b), AugReg Steiner et al. (2021) and DeiT-III Touvron et al. (2022). CKA is a metric that quantifies representational similarity between models, it ranges from 0 (no similarity) to 1 (perfect similarity). We categorize teacher pairs into three regimes: (i) low similarity (CKA 0.2), (ii) moderate similarity ( CKA 0.5), and (iii) high similarity (CKA 0.7). Our findings suggest that self-supervised, regularized multi-teacher distillation performs best when teachers exhibit moderate similarity, i.e., when CKA 0.5. We present these results in Table 8. We color code our models as green if they surpass their teachers and red if they don't. We observe that partial alignment of representations yields the best results, as excessive similarity leads to redundant information, while insufficient similarity introduces conflicting signals. We present additional results of classification, image retrieval, transfer learning, video segmentation and results on other ImageNet dataset for iBoT+AugReg and MUGS+AugReg multi-teacher distillation with DINO self-supervision for SeFA in Appendix C.1.

**Increasing the student size** For our main results, we use a ViT-S/16 encoder. However, in Table 5, we report results using a ViT-B/16 instead. All approaches are trained for 200 epochs following the same experimental setup as described in Section 4 with a ViT-B/16 instead of a ViT-S/16. Consistent with earlier observations, our method yields a significantly stronger pre-trained model using self-supervision regulation for multi-teacher distillation than the existing state-of-the-art method UNIC Sariyildiz et al. (2024) without the self-supervised regularization.[1]

**Additional Results, Analysis, and Protocols.** Further implementation details are provided in our codebase[2] and in Appendix A. We provide detailed evaluation protocols for classification, image retrieval, and segmentation in Appendix B. Appendix C presents further results, including experiments with varying numbers of teachers in multi-teacher distillation, SeFA benchmarks across classification, image retrieval, and segmentation tasks, and comparisons of (AugReg + MUGS) and (AugReg + iBOT) under DINO self-supervision with multi-teacher distillation. We further extend the discussion on changing teacher combinations in Appendix C.1, considering teachers trained on datasets beyond the ImageNet family and for ViT-B/16. Distilling a student model with the same capacity as the teachers is considerably more challenging than distilling a smaller student using multiple teachers. For our main results, we focus on the former scenario, with the latter explored in more detail in Appendix C.4. Additionally, we provide ImageNet-1k validation classification accuracy comparisons of SeFA with off-the-shelf models pre-trained using multi-teacher distillation in Appendix C.8. We observe that SeFA outperforms the state-of-the-art by 1.54% on ImageNet classification while utilizing the fewest student and teacher parameters. The relationship between self-supervision and distillation is further discussed in Appendix D, and the limitations of SeFA are outlined in Appendix E. Appendix F and G provides a qualitative analysis of the representation space, attention maps, comparing SeFA with its DINO and AugReg teachers, as well as SeFA models trained with different teacher combinations for ImageNet and real world images.

## 6 CONCLUSION

In this work, we introduced Self-supervised Feature Aggregation (SeFA), a novel framework that advances the state of the art in multi-teacher distillation for vision models. As shown in Section 5, SeFA goes beyond the traditional scenario of distilling a smaller student from a larger teacher: **SeFA enables multi-teacher distillation in which the student can outperform its teachers** across a wide range of computer vision tasks. SeFA transfers effectively without task-specific training, generalizes to out-of-distribution data, and offers strong retrieval, demonstrating its ability to learn transferable, domain-agnostic, and robust representations beyond its training setup. While SeFA achieves competitive performance even with arbitrary teacher combinations, this property of surpassing teachers is observed only when the teachers exhibit a linear CKA of approximately 0.5. Consolidating results across a variety of student and teacher combinations, we find that SeFA consistently establishes new state-of-the-art performance for multi-teacher distillation.

---

[1]We observe that, for image retrieval, the performance of ViT-B/16 drops compared to ViT-S/16, as ViT-B/16 tends to overfit on datasets smaller than 12 million images Alabdulmohsin et al. (2022).

[2]We will open-source our code upon acceptance of SeFA.

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

## A    ADDITIONAL IMPLEMENTATION DETAILS

In Fig. 1, we provide a schematic illustration of SeFA. In Box A, we depict the multi-teacher distillation setup employed in SeFA. Here, $Aug_1$ and $Aug_2$ denote teacher-specific augmentations used to ensure proper preprocessing of images before feature extraction from the teachers. The preprocessing follows the official implementation of each respective teacher. $dh_1$ and $dh_2$ represent two distinct distillation heads that map the student representation (logits) $u_s$ to each teacher's representation (logits) space. Structurally, these heads are similar to the self-supervised heads used in DINO Caron et al. (2021). In Box B of the same figure, we illustrate the self-supervised component of our approach. Since Fig. 1 depicts SeFA with DINO Caron et al. (2021), Box B corresponds to DINO self-supervision. Here, $t$ and $t'$ represent differential augmentations sampled from a distribution $T$, and $sh_1$ and $sh_2$ are the two self-supervised heads as described in Caron et al. (2021).

## B    EVALUATION PROTOCOLS

### B.1    EVALUATION ON OTHER IMAGENET DATASETS

ImageNet-A Hendrycks et al. (2021b), ImageNet-C Hendrycks & Dietterich (2019), ImageNet-R Hendrycks et al. (2021a), and ImageNetV2 Recht et al. (2019)

The ImageNet-V2 dataset Recht et al. (2019) was introduced to address the absence of a dedicated test split in ImageNet and to quantify overfitting with respect to the original ImageNet validation set Touvron et al. (2019; 2022). We select the best-performing linear probe from Section 5 and evaluate it on ImageNet-V2 to assess generalization to a matched but independently collected distribution.

In addition, we evaluate on the following challenging robustness benchmarks:

- **ImageNet-A** Hendrycks et al. (2021b): Contains naturally occurring, unmodified adversarial examples that are easily misclassified by standard ImageNet-trained models.
- **ImageNet-C** Hendrycks & Dietterich (2019): Applies algorithmically generated corruptions (e.g., noise, blur, weather, and digital distortions) to ImageNet images, testing corruption robustness across severity levels.
- **ImageNet-R** Hendrycks et al. (2021a): Replaces natural photographs with artistic renditions, cartoons, and other non-photographic styles, evaluating robustness to substantial distribution shifts in texture and style.

Together, these benchmarks provide complementary perspectives on model generalization and robustness under distribution shifts, natural adversarial inputs, and input corruptions.

### B.2    TRANSFER LEARNING

To evaluate the transferability of pre-trained features, we follow previous work Caron et al. (2021); Oquab et al. (2023); Sariyildiz et al. (2024); Zhou et al. (2022a); Chen* et al. (2021) and train a linear classifier on popular transfer datasets such as CIFAR-10 and CIFAR-100 Krizhevsky et al. (2009), Oxford Flowers-102 Nilsback & Zisserman (2008), and Oxford-IIIT Pets Parkhi et al. (2012). These datasets are particularly important for transfer learning evaluation because:

- **CIFAR-10/100**: Provide standardized benchmarks for general object recognition with 10 and 100 classes respectively. Their difficulty stems from low-resolution ($32\times32$) images and significant intra-class variation, testing the encoder's ability to extract discriminative features from limited visual information.
- **Oxford Flowers-102**: Challenges models with fine-grained classification among 102 flower species, where subtle visual differences between classes require highly discriminative features. The dataset's long-tailed distribution and limited samples per class (minimum 40 images) add to the difficulty.
- **Oxford-IIIT Pets**: Evaluates performance on fine-grained pet recognition with 37 categories, complicated by varying animal poses, lighting conditions, and occlusions. The need to

distinguish between similar breeds makes this particularly challenging for feature quality assessment.

We use a frozen pre-trained encoder, and train a linear classifier on top of feature outputs from the frozen encoder and a learning rate sweep across 45 values in $[10, 10^{-4}]$.

### B.3 DENSE PREDICTION TASKS

To understand the dense prediction capabilities of our ImageNet-pretrained model, we extend evaluation to the Densely Annotated VIdeo Segmentation (DAVIS) dataset Pont-Tuset et al. (2017). This benchmark is critically important for assessing:

- **Spatiotemporal Feature Transfer**: DAVIS tests whether image-trained features generalize to video understanding, requiring *temporal consistency* in segmentation across frames—a key challenge absent in static image tasks.
- **Fine-grained Semantics**: The dataset's multiple annotated objects per sequence (average 2.3 per frame) and fine structures (e.g., animal fur, thin objects) demand high-resolution feature discrimination, pushing the limits of pretrained encoders.
- **Dynamic Scene Robustness**: Challenges like fast motion (average 24.5px/frame displacement) and severe inter-object occlusion Li et al. (2017) stress-test the model's ability to maintain object identity and boundaries under deformation.

The task involves segmenting specific objects via nearest-neighbor retrieval, which is more challenging than image segmentation beacuse in some cases DAVIS objects occupy a very small frame area, exacerbating feature localization demands. Furthermore a lot of frames contain occlusions, requiring features to disentangle semantics.

Following Caron et al. (2021); Zhou et al. (2022a); Su & Ji (2024), we report three metrics:

- **Region similarity** $\mathcal{J}_{\mathrm{m}}$: Measures IoU under motion, sensitive to temporal drift.
- **Contour accuracy** $\mathcal{F}_{\mathrm{m}}$: Evaluates boundary precision for thin structures.
- **Combined** $(\mathcal{J}\&\mathcal{F})_{\mathrm{m}}$: Balanced assessment of holistic segmentation quality.

### B.4 IMAGE RETRIEVAL

"In Image Retrieval, the task is to return a ranked list of database images most similar to a given query, typically involving fine-grained landmark instance recognition using frozen pre-trained features. The revisited Oxford and Paris datasets Radenović et al. (2018) are critical for evaluating **generalization under real-world variations**, with challenges including:

- **Viewpoint/lighting shifts** (Hard split contains 60% more extreme variations than Medium)
- **Partial occlusions and crop artifacts**, testing feature robustness
- **Fine-grained distinctions** (e.g., different facade details in Paris buildings)

Following Caron et al. (2021); Zhou et al. (2022a), we report **Mean Average Precision (mAP)** on Medium (M) and Hard (H) splits.

.

## C  ADDITIONAL RESULTS

### C.1  CHANGING TEACHER COMBINATION FOR ViT B/16

In Section 5.2, we analyzed the effect of varying teachers for ViT-S/16 in SeFA. We observed that all teachers contributed useful features, enabling the student to achieve competitive performance. Notably, moderately aligned teachers facilitated multi-teacher distillation, allowing the SeFA-trained student to outperform the teachers used during pre-training. In this section, we extend this analysis to

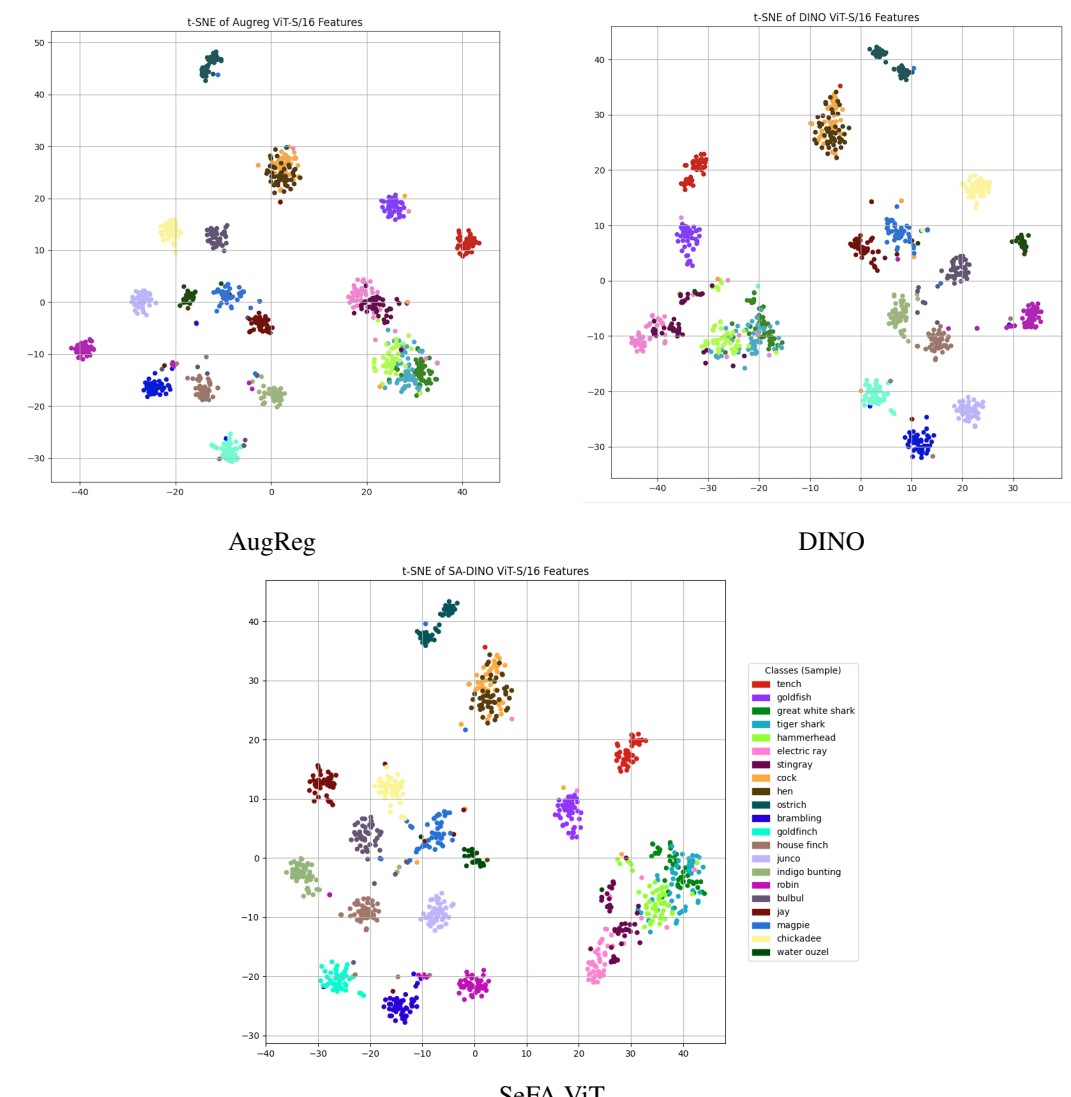

AugReg

DINO

SeFA ViT

Figure 2: t-SNE visualizations for AugReg, DINO (top row), and SeFA ViT (bottom row). We observe that some clusters occupy similar positions across all three representation spaces (e.g., the orange and dark brown clusters). In contrast, other clusters, such as the blue and dark green clusters, show that SeFA is more closely aligned with AugReg.

ViT-B/16. Similar to Table 8, Table 9 presents results for ViT-B/16 students distilled using teachers of the same capacity. Consistent with the findings in Section 5.2, moderately aligned teachers yield better student performance than fully aligned or poorly aligned teachers. Moreover, we identify that the optimal range of alignment lies between 0.50 and 0.54 on the linear CKA scale.

## C.2 IMPROVING ADVERSARIAL CLASSIFICATION PERFORMANCE ON IMAGENET-A

In Section 5, we demonstrated that SeFA-trained models outperform both strongly supervised and self-supervised teachers across most benchmarks, with the exception of ImageNet-A. We now extend this analysis by evaluating the performance of a pre-trained student model under a linear probing followed by finetuning (LP-FT) protocol. Specifically, we first train a linear probe on top of the frozen backbone, then select the best-performing probe and unfreeze the backbone for end-to-end finetuning over 100 epochs. This LP-FT strategy preserves the pre-trained representation space of the student while enabling more effective adaptation Kumar et al. (2022). We additionally benchmark

| Teacher Combination | Pre-training + *Linear* 200 PTE | CKA | Similarity | In1k Acc.(%) |
|---|---|---|---|---|
| DINO + CLIP[1] | 79.30 | 0.5915 | **High** | 79.76 |
| DINO + SigLIP[2] | 80.14 | 0.5705 | **High** | 82.57 |
| DINO + AugReg | 80.71 | 0.5401 | **Moderate** | 79.70 |
| DINO + DeiT-III | 81.85 | 0.3987 | **Low** | 83.60 |
| DINO + CLIP[3] | 80.30 | 0.3313 | **Low** | 85.73 |

Table 9: Top-1 Accuracy on the ImageNet-1K validation dataset using SeFA ViT-B/16 models distilled from teacher pairs. Cells are green when student accuracy exceeds the best teacher, and red otherwise. CKA values quantify representational similarity, with regimes defined as **Low** (CKA ≈ 0.2), **Moderate** ( CKA ≈ 0.5), and **High** (CKA ≈ 0.8). Here, CLIP[1] represents CLIP Radford et al. (2021) train on the LAION5B dataset Beaumont (2022), SigLIP[2] represents SigLIP Zhai et al. (2023) train on the WeBLI dataset Chen et al. (2022), and CLIP[3] represents CLIP Radford et al. (2021) train on the YFCC dataset Thomee et al. (2016)

these LP-FT models on ImageNet-A. As shown in Table 10, LP-FT consistently enhances robustness: SeFA not only achieves higher accuracy on ImageNet-1k but also surpasses the best-performing teacher on ImageNet-A by approximately 3%. These results underscore that end-to-end finetuning further strengthens SeFA's robustness to adversarial noise.

| Model | IN-1k | IN-A |
|---|---|---|
| **Teacher models** | | |
| AugReg | 76.50 | 16.83 |
| **SeFA models** | | |
| AugReg + DINO | 79.96 | 19.85 |
| AugReg + MUGS | 79.92 | 19.43 |
| AugReg + iBOT | 80.20 | 19.87 |

Table 10: Comparison of teacher and SeFA models with the ViT-S/16 architecture after end-to-end finetuning on ImageNet-1k, followed by evaluation of the finetuned models on ImageNet-A.

### C.3 SCALING NUMBER OF TEACHERS

In Section 5, we study knowledge distillation using only two teachers. Here, we expand from two to three teachers by incorporating ViT S/16 AugReg alongside ViT S/16 iBOT and DINO. We observe that iBOT and DINO teachers exhibit very high CKA values (≈ 0.8), while AugReg shows moderate similarity with both DINO and iBOT (≈ 0.5).

Our key findings reveal that: (1) when teachers have moderate similarity (AugReg+DINO or AugReg+iBOT), the student surpasses its teachers; (2) with highly similar teachers (DINO+iBOT), the student underperforms relative to the teachers; but (3) adding a moderately-aligned teacher (AugReg) to highly-similar pairs stabilizes the student and improves representation learning.

Notably, while the three-teacher combination (AugReg+iBOT+DINO) yields a student that improves over all individual teachers, it does not outperform the student trained solely on iBOT and AugReg.

### C.4 SCALING TEACHER CAPACITY

Distilling a student model with the same capacity as the teachers is considerably more challenging than distilling a smaller student using multiple teachers. We have already established SeFA's state-of-the-art performance over existing approaches in Section 5, Appendix C.5, and Appendix C.6. In Table 12, we report results for distilling a smaller student from larger teacher models. Specifically, SeFA ViT-Small trained using ViT-Base teachers outperforms the equivalent SeFA ViT-Small trained using ViT-Small teachers by approximately 0.35%.

| Model | Linear | Pre-training + Linear | |
|---|---|---|---|
| | | 100 PTE | 200 PTE |
| *Teacher Models* | | | |
| DINO ViT-S/16[t1] | 76.09 | - | - |
| iBOT ViT-S/16[t2] | 77.05 | - | - |
| AugReg ViT-S/16[t3] | 76.50 | - | - |
| SeFA ViT-S/16 (*t1 + t2*) | - | 76.57 | 76.75 |
| SeFA ViT-S/16 (*t1 + t3*) | - | 77.80 | 77.92 |
| SeFA ViT-S/16 (*t2 + t3*) | - | 78.12 | 78.34 |
| SeFA ViT-S/16 (*t1 + t2 + t3*) | - | 77.96 | 78.1 |

Table 11: Top-1 accuracy on the ImageNet-1k validation set for SeFA-based multi-teacher distillation using multiple teachers. We observe that the degree of alignment between teachers is more important than the sheer number of teachers: two moderately aligned teachers outperform three teachers of the same model size.

| SeFA Student | Teachers | In1k Acc (%) |
|---|---|---|
| ViT Tiny/16 | DINO (ViT Small/16) AugReg (ViT Small/16) | 71.91 |
| ViT Small/16 | DINO (ViT Base/16) AugReg (ViT Base/16) | 78.27 |
| ViT Base/16 | iBOT (ViT Large/16) AugReg (ViT Large/16) | 81.68 |

Table 12: Top-1 Accuracy on the ImageNet-1K validation dataset. For each SeFA model mentioned we use the DINO self-supervised objective with the multi-teacher distillation loss for 200 epochs of pre-training followed by linear probing for 100 epochs.

To further understand the significance of the results for ViT-Tiny, we compare it with UNIC, the baseline DINO, and the state-of-the-art self-supervised approach for small architectures without distillation—SSLight Tan et al. (2023)—in Table 13. We observe that SeFA-trained ViT-Tiny substantially outperforms the comparable multi-teacher distillation method UNIC. Moreover, it achieves a notable improvement of 2.41% over SSLightTan et al. (2023). These results further demonstrate SeFA's effectiveness in training small-capacity student models relative to their teachers.

| Approach | Student Arch. | Teachers | In1k Acc (%) |
|---|---|---|---|
| DINO Caron et al. (2021) | ViT Tiny/16 | - | 66.7 |
| DINO SSLight Tan et al. (2023) | ViT Tiny/16 | - | 69.5 |
| UNIC Sariyildiz et al. (2024) | ViT Tiny/16 | DINO (ViT Small/16) AugReg (ViT Small/16) | 61.89 |
| SeFA | ViT Tiny/16 | DINO (ViT Small/16) AugReg (ViT Small/16) | **71.91** |

Table 13: Top-1 Accuracy on the ImageNet-1K validation dataset for ViT Tiny/16 backbone. For all experiments the student has been pre-trained for 200 epochs, followed by linear probing.

## C.5 IBOT+AUGREG WITH DINO

From Table 8, we observe that teacher pairings with moderate alignment—(DINO + AugReg), (iBOT + AugReg), and (MUGS + AugReg)—yield students that surpass their teachers. While Section 5 presents results for (DINO + AugReg), here and in Section C.6 we focus on (iBOT + AugReg) and (MUGS + AugReg) using multi-teacher distillation combined with DINO self-supervised loss (ViT S/16).

For the iBOT + AugReg combination, we find that—similar to DINO + AugReg—the student consistently outperforms both teachers across all ImageNet and transfer learning benchmarks, except for ImageNet-A. We observe that end-to-end finetuning considerably improves ImageNet-A performance as discussed in Appendix C.2.Tables 15 and 16 show results for image retrieval and video segmentation, respectively. In retrieval Table 15, the student fails to surpass either teacher, likely due to iBOT's weaker baseline performance compared to DINO. However, for video segmentation, the student consistently outperforms both teachers.

| **Model** | INt-r | IN-A | IN-C ($\downarrow$) | IN V2 | Transfer |
|---|---|---|---|---|---|
| *Teacher Models* | | | | | |
| iBOT | 36.83 | 12.53 | 64.32 | 73.43 | 85.40 |
| Augreg. | 35.73 | 16.83 | 68.69 | 71.77 | 86.85 |
| Best Teacher | 36.83 | **16.83** | 64.32 | 73.43 | 86.85 |
| SeFA (*ours*) | **38.85** | 12.27 | **63.26** | **74.57** | **87.15** |
| UNIC | 28.42 | 6.64 | 76.27 | 66.12 | 81.95 |

Table 14: iBoT + AugReg Teacher using DINO

| **Pretrain** | $\mathcal{R}_{Oxford}$ | | $\mathcal{R}_{Paris}$ | |
|---|---|---|---|---|
| | M | H | M | H |
| Augreg | 28.94 | 9.71 | 63.41 | 38.24 |
| iBOT | 33.55 | 10.55 | 58.47 | 31.23 |
| Best Teacher | 33.55 | 10.55 | **63.41** | **38.24** |
| SeFA | **38.22** | **14.49** | 63.03 | 35.53 |
| UNIC | 29.24 | 8.86 | 57.2 | 28.86 |

| Model | $\mathcal{J}_\mathrm{m}$ | $\mathcal{F}_\mathrm{m}$ | $(\mathcal{J}\&\mathcal{F})_\mathrm{m}$ |
|---|---|---|---|
| *Teacher Models* | | | |
| iBOT | 60.05 | 62.51 | 61.28 |
| Augreg | 47.71 | 49.38 | 48.54 |
| Best teacher | 60.05 | 62.51 | 61.28 |
| SeFA (ours) | **60.09** | **63.56** | **61.82** |
| UNIC | 52.77 | 55.49 | 54.13 |

Table 15: Comparison of pretraining methods on ROx and RPar under Medium (M) and Hard (H) protocols. iBoT + AugReg Teacher using DINO.

Table 16: Mean region similarity $\mathcal{J}_\mathrm{m}$, contour accuracy $\mathcal{F}_\mathrm{m}$, and combined $(\mathcal{J}\&\mathcal{F})_\mathrm{m}$ on DAVIS 2017.iBoT + AugReg Teacher using DINO.

## C.6 MUGS+AUGREG WITH DINO

In this section, we evaluate the student model trained with MUGS + AugReg teachers using DINO self-supervised loss and multi-teacher distillation. We assess performance across three tasks, with results shown in:

- Table 17 for image classification
- Table 18 for image retrieval
- Table 19 for video segmentation

Consistent with the (DINO + AugReg) and (iBoT + AugReg) teacher pairs, our analysis reveals two key findings:

- The student achieves superior classification performance on all benchmarks except ImageNet-A. End-to-end finetuning as described in Appendix C.2, improves ImageNet-A classification of SeFA distilled from Augreg and iBOT to 19.87%.
- For video segmentation, it consistently outperforms both teachers.

However, in image retrieval—similar to the (iBOT + AugReg) case—the student only surpasses its teachers on the Revisited-Oxford dataset. This limitation stems from MUGS's weaker baseline retrieval performance, analogous to iBOT's subpar retrieval capability compared to DINO.

| Model | IN-R | IN-A | IN-C ($\downarrow$) | IN V2 | Transfer |
|---|---|---|---|---|---|
| *Teacher Models* | | | | | |
| MUGS ViT-S/16 | 35.67 | 9.79 | 65.32 | 70.84 | 86.47 |
| Augreg. ViT-S/16 | 35.73 | 16.83 | 68.69 | 71.77 | 86.85 |
| Best Teacher | 35.73 | **16.83** | 65.32 | 71.77 | 86.85 |
| SeFA ViT-S/16 (*ours*) | **38.56** | 10.55 | **63.79** | **73.80** | **87.35** |

Table 17: Mugs + AugReg Teacher using DINO

## C.7 CHANGING $\lambda_1$ IN LOSS EQUATION 4

From Equation 4, the coefficient $\lambda_1$ controls the relative importance of multi-teacher distillation versus DINO self-supervision. In Table 20, we ablate different values of $\lambda_1$ to study this trade-off. We observe that accuracy degrades when $\lambda_1$ is set either too high or too low. A large $\lambda_1$ emphasizes the self-supervised objective, reducing the influence of the multi-teacher signal and limiting the ability of the student to distill useful priors. Conversely, a small $\lambda_1$ overemphasizes distillation, encouraging the student to mimic teachers without sufficient filtering, which leads to diverse but less discriminative features.

## C.8 COMPARISON WITH ARBITRARY MODELS

We provide a comparison with contemporary knowledge distillation approaches in Appendix Table 22, focusing on methods that use a comparable ViT-B/16 student architecture. It is important to note the key differences in training: AM-RADIO Ranzinger et al. (2024) utilizes a custom ViT-B/16 variant, while SAK Lu et al. (2025) introduces a mixture of experts via a trainable router to combine teachers. With the exception of AM-RADIO, all other baselines are trained on ImageNet-1K, often leveraging very large teacher networks. Our key finding is that SeFa, despite being distilled from a combination of smaller and more efficient teachers, achieves superior performance. This result underscores a significant advantage of our selective feature aggregation strategy. We also scale SeFA's teachers from two ViT-B to ViT-L networks. We observe that SeFA improves further by 1% on IN-1k validation.

## C.9 BACKGROUND DEPENDENCE

Deep neural network for visual tasks rely on both foreground objects and image backgrounds. Even when the correct foreground object is present, such models often make incorrect predictions when the image background is changed, and they are especially vulnerable to adversarially chosen backgrounds Xiao et al. (2020). To systematically study this background reliance, we utilize the ImageNet-9 (IN9) dataset, which includes nine coarse-grained classes and seven variants created by mixing foregrounds and backgrounds from different images. Four of these variants—Only-FG (O.F.), Mixed-Same (M.S.), Mixed-Rand (M.R.), and Mixed-Next (M.N.)—retain the original foreground while modifying the background. The remaining three—No-FG (N.F.), Only-BG-B (O.BB.), and Only-BG-T (O.BT.)—mask the foreground entirely. To quantify a model's dependence on background signals, we use the BG-GAP metric, defined as the accuracy difference between the MIXED-SAME and MIXED-RAND variants. A smaller BG-GAP indicates reduced reliance on background information for correct predictions, which is desirable for robust object recognition. As shown in Table 21,

| Method | $\mathcal{R}_{Oxford}$ | | $\mathcal{R}_{Paris}$ | |
|---|---|---|---|---|
| | M | H | M | H |
| AugReg | 28.94 | 9.71 | 63.41 | 38.24 |
| MUGS | 32.35 | 10.35 | 58.35 | 30.48 |
| Best Teacher | 32.35 | 10.35 | **63.41** | **38.24** |
| SeFA-DINO | **35.84** | **11.84** | 61.65 | 33.33 |

Table 18: Image retrieval mAP (%) on ROxford (ROx) and RParis (RPar) under Medium (M) and Hard (H) evaluation protocols.

| Model | $\mathcal{J}_{\mathrm{m}}$ | $\mathcal{F}_{\mathrm{m}}$ | $(\mathcal{J}\&\mathcal{F})_{\mathrm{m}}$ |
|---|---|---|---|
| MUGS ViT-S/16 | 58.92 | 62.85 | 60.88 |
| AugReg ViT-S/16 | 47.71 | 49.38 | 48.54 |
| Best Teacher | 58.92 | 62.85 | 60.88 |
| SeFA ViT-S/16 (ours) | **60.05** | **63.44** | **61.74** |

Table 19: Video segmentation performance on DAVIS 2017 showing mean region similarity ($\mathcal{J}_{\mathrm{m}}$), contour accuracy ($\mathcal{F}_{\mathrm{m}}$), and their combination.

| $\lambda_1$ | Accuracy (%) |
|---|---|
| 0.30 | 77.32 |
| 0.40 | 77.61 |
| 0.50 | **77.80** |
| 0.60 | 77.50 |
| 0.65 | 77.31 |
| 0.70 | 77.04 |

Table 20: Effect of varying $\lambda_1$ on ImageNet-1K accuracy for SeFA using AugReg and DINO teachers with DINO self-supervision and multi-teacher distillation trained for 100 epochs using ViT S/16.

we observe that multi-teacher distillation, when regularized by self-supervision (SeFA), results in the smallest accuracy drop across background variants, highlighting its effectiveness in mitigating background sensitivity. We further analyze SeFa in Appendix G.

| Metric | DINO | AugReg | UNIC | SeFA ViT |
|---|---|---|---|---|
| original | 96.00% | 96.25% | 93.83% | **96.52%** |
| mixed_same | **89.33%** | 85.75% | 80.57% | 89.14% |
| only_fg | 89.38% | 84.99% | 74.89% | **89.78%** |
| mixed_rand | 81.38% | 78.00% | 69.56% | **81.56%** |
| mixed_next | **79.40%** | 76.17% | 66.20% | 79.38% |
| no_fg | 51.60% | **53.06%** | 45.80% | 52.96% |
| only_bg_b | **22.47%** | 21.98% | 15.28% | 19.36% |
| only_bg_t | **17.63%** | 17.16% | 15.11% | 16.07% |
| BG-gap ($\downarrow$) | 7.95% | 7.75% | 11.01% | **7.58%** |

Table 21: ImageNet-9 Benchmark Results (ViT-S/16 Variants). All metrics represent accuracy.

## D  SELF-SUPERVISION AND KNOWLEDGE DISTILLATION

Knowledge distillation plays a central role in many state-of-the-art self-supervised learning frameworks, particularly through the mechanism of self-distillation. In this setup, a student model learns from a teacher model, which is periodically updated as an exponential moving average of the student. Between updates, both models are trained to produce consistent predictions across different augmentations of the same input. This framework not only encourages the emergence of explicit clustering structures but also allows fine-grained control over the clustering behavior via parameters such as the number of centroids and temperature settings—ultimately facilitating the formation of semantically meaningful image abstractions.

However, popular self-distillation approaches like DINO Caron et al. (2021) and SimSiam Chen & He (2020) operate primarily at the image level, thereby limiting their ability to capture fine-grained semantics. To address this limitation, more recent work such as iBOT Zhou et al. (2022a) and DINOV2 Oquab et al. (2023) has incorporated image-level masked image modeling objectives to promote finer granularity in representation learning. From the perspective of the information bottleneck (IB) principle, methods such as DINO and iBOT Zhou et al. (2022a) are susceptible to over-compressionSu & Ji (2024). This compression largely stems from aggressive prediction sharpening between the teacher and student models—a mechanism essential to prevent representational collapse but one that also constrains the information retained in learned features, thus compromising generalization.

To overcome these challenges, recent methods like Unsqueezed Distillation-based Self-Supervision (UDI) Su & Ji (2024) build upon the IB principle with explicit clustering objectives. Similar to the addition to additional [cls] tokens as registers Darcet et al. (2023), UDI introduces an additional class token to circumvent the information bottleneck. This enables the model to preserve richer semantic content. This design offers both improved generalization and enhanced control over the clustering process, advancing the state-of-the-art in self-supervised representation learning.

Use of self-supervision and knowledge has been previously studied by In Xu et al. (2020), the authors train a large teacher in a supervised manner, followed by self-supervised distillation to a smaller student. They argue that self-supervision is largely ineffective for smaller architectures, motivating the use of a single larger teacher for distillation. In contrast, SeFA enables distillation from pre-trained architectures into students of the same capacity. Furthermore, SeFA optimizes a multi-task loss that jointly combines self-supervision and distillation, whereas Xu et al. (2020) employs a two-step process, applying one stage for pure supervision and another for self-supervision.

In addition to the approaches discussed in Section 2, other methods, such as Li et al. (2024), perform multi-teacher distillation; however, instead of using pre-trained teacher models, they use differently parameterized copies of the student. This approach extends the self-distillation paradigm employed in state-of-the-art self-supervised methods, including DINO Caron et al. (2021), iBOT Zhou et al. (2022a), and UDI Su & Ji (2024), among others.

## E  LIMITATIONS

Although SeFA improves performance beyond that of the teachers used during pretraining, this improvement occurs only when there is a moderate degree of alignment between teachers (linear CKA 0.50–0.54). In cases of very high or very low similarity, SeFA falls short of matching the teacher's performance. Achieving a "one model for all tasks" would require the model to actively identify and refine only globally useful features, independent of the alignment between teachers. Additionally, due to computational constraints, we train self-supervised approaches with a relatively small batch size (128 images per GPU). Self-supervised methods generally benefit from larger batch sizes; consequently, further performance gains could likely be achieved with sufficiently large batches.

## F  ATTENTION MAPS

### F.1  QUALITATIVE ANALYSIS OF ATTENTION MAPS ACROSS AUGREG, DINO, AND SEFA

We provide a comparative analysis of attention maps for three models: AugReg, DINO, and SeFA (a student distilled from AugReg and DINO). For each attention map, we present (from left to right) the

| Model | Params | Teacher Params | Data | Teachers | Linear |
|-------|--------|----------------|------|----------|--------|
| AM-RADIO ViT-B/16 Ranzinger et al. (2024) | 118M | ~3.2B | DataComp-1B | DFN CLIP ViT-H/14 SigLIP 400M DINOv2-g-reg SAM-H | 78.24 |
| THEIA ViT-B/16* Shang et al. (2024) | 86M | ~1.23B | IN-1K | DINOv2 ViT-L/14 ViT-H/14 CLIP ViT-L/16 | 75.2 |
| THEIA ViT-B/16 Shang et al. (2024) | 86M | ~1.23B | IN-1K | DINOv2 ViT-L/14 ViT-H/14 CLIP ViT-L/16 | 72.1 |
| SAK ViT-B/16 Lu et al. (2025) | 134M | ~0.26B | IN-1K | DINOV2 ViT-B/16 SAM ViT-B/16 CLIP ViT-L/16 | 79.16 |
| SeFA ViT-S/16 *(ours)* | 22M | ~0.17B | IN-1K | DINO ViT-B/16 AugReg ViT-B/16 | 78.26 |
| SeFA ViT-B/16 *(ours)* | 86M | ~0.17B | IN-1K | DINO ViT-B/16 AugReg ViT-B/16 | 80.70 |
| SeFA ViT-B/16 *(ours)* | 86M | ~0.61B | IN-1K | iBOT ViT-L/16 AugReg ViT-L/16 | **81.68** |

Table 22: Comparison of ImageNet-1k accuracy after linear probing between AM-RADIO, THEIA, SAK and SeFA ViT-B/16, including pretraining data and total teacher parameters.

original image, followed by clustering of the attention map tokens, and then the representations from all attention heads. Attention maps are visualized across multiple heads in the last transformer block, highlighting the regions each model focuses on during inference.

### F.1.1 AUGREG: SPARSE, LOCAL DISCRIMINATIVE FOCUS

- **Fish Image (Rows 1 and 2 of Fig. 3)**: AugReg primarily attends to the spine of the fish, ignoring the human subject and the broader context.

- **Cat (Row 3 of Fig. 3)**: Sparse activations are seen around the cat's forehead and eyes, with no coverage of the full object.

- **Wheat Stalk (Row 5 of Fig. 4)**: Attention is narrowly focused on the center of the stalk, failing to incorporate contextual cues from the background.

These patterns suggest AugReg learns highly localized, discriminative parts, suitable for classification but less so for tasks requiring holistic understanding.

### F.1.2 DINO: SEMANTIC, OBJECT-CENTRIC REPRESENTATIONS

- **Cat Image (Row 3 of Fig. 3)**: Attention covers the entire head and both eyes, showing semantically grouped attention across relevant features.

- **Meerkat (Row 4 of Fig. 3)**: The full body structure, including the torso and head, is captured, indicating strong object-level understanding.

- **Butterfly (Row 5 of Fig. 3)**: DINO consistently captures both wings and antennae, often with symmetry across heads.

DINO's attention is more structured and holistic, enabling better transfer to downstream tasks such as segmentation and detection.

### F.1.3 SeFA: Multi-Granular Integration from Teachers

- **Fish Image (Rows 1 and 2 of Fig. 3)**: SeFA blends DINO's global coverage of the fish with AugReg's sharper focus on the fins.
- **Butterfly (Row 5 of Fig. 3)**: Both wings are symmetrically captured, similar to DINO, but with finer resolution of local regions.
- **Train (Row 1 of Fig. 4)**: Attends to the front and body edges, combining AugReg's part-specific and DINO's structural awareness.
- **Jumping Person (Row 3 of Fig. 4)**: SeFA highlights multiple body parts (head, arms, legs) with stronger local contrast.
- **Traffic Scene (Row 4 of Fig. 4)**: Attention includes traffic poles, houses, and cars—capturing more than AugReg and refining DINO's broader coverage.

SeFA exhibits multi-scale attention, suggesting successful distillation of complementary features from both teachers. We also present attention maps of (AugReg + DINO) SeFA trained ViT S/16 on real world images in Fig. 7. We observe that SeFA is able to discern fine grained details images.

- **Focus on Salient Objects:** The attention maps consistently exhibit a strong localization on the most distinct features within the images. For instance, attention is clearly concentrated on the two dogs in the first row, the frying pan in the third row, and the branded cup in the sixth row. This suggests the ViT model effectively learns features relevant to **high-information content regions**.
- **Attention Head Specialization:** The sequence of attention maps (representing different attention heads) indicates a degree of functional specialization. Different heads capture varied visual features or structural aspects of the scene. In the case of the streetlight (Row 4), initial heads may capture the overall structure, while subsequent heads may focus intensely on the verticality of the pole or the light source itself.
- **Structure and Contextual Cues:** While focusing on objects, the attention is not strictly confined. Maps often include dimmer, spread-out patterns over the background (e.g., pavement, sky). This suggests the model integrates **contextual information** and scene layout (as seen in Row 2, the outdoor bench scene) alongside object-specific details.
- **Localized Distillation Efficacy:** The sharp, well-delineated nature of the attention masks is a testament to the efficacy of the **SeFA-trained distillation**. The Self-Attention-based Feature Aggregation method appears to successfully transfer a **clean and interpretable localization ability** from the teacher model, leading to robust feature isolation even on challenging, real-world smartphone captures.

### F.1.4 Summary Table of Observations

| Observation | AugReg | DINO | SeFA |
|---|---|---|---|
| Sparse, part-based attention | ✓ | – | ✓ |
| Global object coverage | – | ✓ | ✓ |
| Background/context sensitivity | – | ✓ | ✓ |
| Fine-grain + structure integration | – | – | ✓ |

Table 23: Qualitative summary of attention behavior across models.

### F.1.5 Conclusion

SeFA inherits AugReg's sharp, localized cues and DINO's global, object-centric semantics. This results in diverse and robust representations that generalize well across tasks requiring fine-to-coarse granularity. Visual evidence across varied samples supports the success of multi-teacher distillation in producing semantically rich attention behavior.

## G  FURTHER ANALYSIS OF SEFA

To better understand the synergy between the two teachers—AugReg and DINO ViT-S/16—and the student ViT-S/16 trained with Self-supervised Feature Aggregation (SeFA), we compute Normalized Mutual Information (NMI) and Silhouette scores using features extracted from frozen backbones on the ImageNet-1k validation set. Normalized Mutual Information (NMI) and Silhouette score serve as two fundamental metrics for evaluating clustering quality in the context of representation learning. NMI quantifies the extent to which the clustering structure corresponds to the ground-truth class labels, with higher values reflecting stronger consistency with label assignments and a more discriminative organization of the learned features. The Silhouette score, in contrast, evaluates both the cohesion within clusters and the degree of separation between them by comparing the average intra-cluster distance to the closest inter-cluster distance. A higher Silhouette score therefore indicates that samples are more tightly grouped within their respective clusters and more distinctly separated from neighboring clusters, reflecting clearer boundaries and more compact structure in the feature space learned by the model.

To visualize the feature space of the frozen encoders, we generate t-SNE visualizations on 20,000 images from the validation set for AugReg ViT-S/16, DINO ViT-S/16, and the student model SeFA ViT-S/16. AugReg, which is trained with labels first on ImageNet-21k and subsequently on ImageNet-1k, achieves the highest NMI (0.8114) and Silhouette score (0.0595). Its t-SNE projection reveals tight and well-separated clusters, reflecting strong alignment with class labels—behavior expected from a well-trained supervised neural network. In contrast, DINO, trained in a fully self-supervised manner without access to labels, attains a lower NMI (0.7786) and Silhouette score (0.0502). The corresponding t-SNE visualization exhibits more scattered clusters that are nevertheless semantically organized, capturing meaningful similarities among samples but with looser and less sharply defined class boundaries. The SeFA student model, distilled jointly from AugReg and DINO, displays a hybrid cluster structure. While certain clusters retain the compactness characteristic of AugReg, others exhibit the spread typical of DINO. Its NMI (0.7796) and Silhouette score (0.0499) are closer to DINO's values, suggesting that the student has not simply averaged the representations of its teachers but has instead synthesized complementary strengths from both. Through its non-linear optimization procedure, SeFA achieves a balance between label-consistent discriminability and semantic flexibility, effectively integrating features inherited from AugReg and DINO.

Furthermore, we compute Centered Kernel Alignment (CKA) values on top of frozen backbone features of AugReg, DINO and SeFA ViT-S/16. Centered Kernel Alignment (CKA) is a robust metric for comparing the similarity of feature representations learned by different neural networks. It quantifies how structurally aligned two sets of activations are, even if they differ in dimensionality or undergo transformations like rotation or scaling. Unlike simpler measures, CKA is invariant to such transformations and provides a normalized score between 0 and 1, making it well-suited for analyzing representational similarity across models, layers, or training settings. This makes it especially valuable in settings like model distillation, where understanding how closely a student mimics its teachers is crucial.

The CKA values (Centered Kernel Alignment) reveal the structural relationships between the feature representations of the models. The student model SeFA exhibits a very high CKA score with DINO (0.8744), indicating that its learned representations are strongly aligned with those of DINO. In contrast, its similarity with AugReg is moderate (0.5734), suggesting that while SeFA incorporates some class discriminative structure from AugReg, it is far more influenced by the semantic feature space of DINO. Furthermore, the relatively low CKA between DINO and AugReg (0.5082) highlights the fundamental difference in their representational structures: DINO's self-supervised semantic organization contrasts with AugReg's label-driven and class-specific characteristics. These relationships confirm that SeFA is not simply blending its teachers but is instead leaning heavily toward DINO, structurally aligning its features with DINO's while partially retaining some supervised traits from AugReg.

| CKA | SeFA | AugReg |
|-----|------|--------|
| AugReg | 0.5734 | 1.0 |
| DINO | 0.8744 | 0.5082 |

Table 24: CKA similarity scores between teacher models.

| Model | NMI | Silhouette Score |
|-------|-----|------------------|
| SeFA | 0.7796 | 0.0449 |
| AugReg | **0.8114** | **0.0595** |
| DINO | 0.7786 | 0.0502 |

Table 25: Comparison of NMI and Silhouette scores across models.

### G.1 COMPARING STUDENT TRAINED WITH TEACHER COMBINATIONS

#### G.1.1 PERCEPTUAL DIFFERENCES ACROSS EXAMPLES

We highlight qualitative differences across a range of visual concepts. Below are observations on specific examples from the attention visualizations:

- **Dog Running (Row 1 of Fig. 5)**: MUGS shows stable focus across heads (e.g., ears, legs, eyes), DINO covers the entire foreground with some irrelevant patches, and iBOT varies in quality — some heads capture facial regions, others are diffuse.

- **Two White Dogs (Row 2 of Fig. 5)**: MUGS provides precise attention centered on the dogs, including their heads and body contours. DINO activates large background areas. iBOT focuses on the dogs but has noisy, inconsistent attention across heads.

- **Flower with Insect (Row 3 of Fig. 5))**: DINO captures both the insect and the leaf but lacks spatial focus, while MUGS produces tight attention and clustering over the insect. iBOT detects the insect but exhibits noisy heads with less confident boundaries.

- **Train (Row 4 of Fig. 5)**: DINO attends to both the train and tracks, while MUGS tightly segments the nose and front. iBOT shows partial attention, but heads often ignore key structural elements.

- **White Van (Row 5 of Fig. 5)**: MUGS produces structured patch groups for headlights and grill. DINO distributes attention over the background and wheels. iBOT often fails to focus centrally, with heads highlighting disconnected regions.

- **Two brown Dogs (Row 1 of Fig. 6)**: MUGS exhibits clear attention on both cubs. DINO's attention is split and background-heavy. iBOT shows mid-level focus but lacks fine localization.

- **Clock and Bottle (Row 2 of Fig. 6)**: MUGS clearly separates the circular clock and the vertical bottle in both patch segmentation and attention heads. DINO's attention is diffuse over the wall texture. iBOT shows partial coverage but with less structural awareness.

- **Green Plant (Row 3 of Fig. 6)**: MUGS attention is tightly aligned with the butterfly and flower, producing clean clusters. DINO spreads attention across the leaves. iBOT captures key parts but misses finer details.

**Summary of Observations**    Table 26 summarizes what each distilled model captures well and what it tends to miss.

Table 26: Qualitative comparison of distilled models trained with AugReg.

| Model | Captures Well | Misses |
|-------|---------------|--------|
| **AugReg+DINO** | Context and object jointly; diverse multi-head perspectives | Less spatial compactness; noisy patch clustering |
| **AugReg+MUGS** | Fine object boundaries; highly semantic patch grouping; consistent across heads | Less emphasis on background context |
| **AugReg+iBOT** | Moderate object focus; some discriminative attention patterns | Inconsistent heads; weaker boundary precision; coarse clusters |

## H  LLM USAGE

LLMs have so far been used primarily for cleaning and improving textual content.

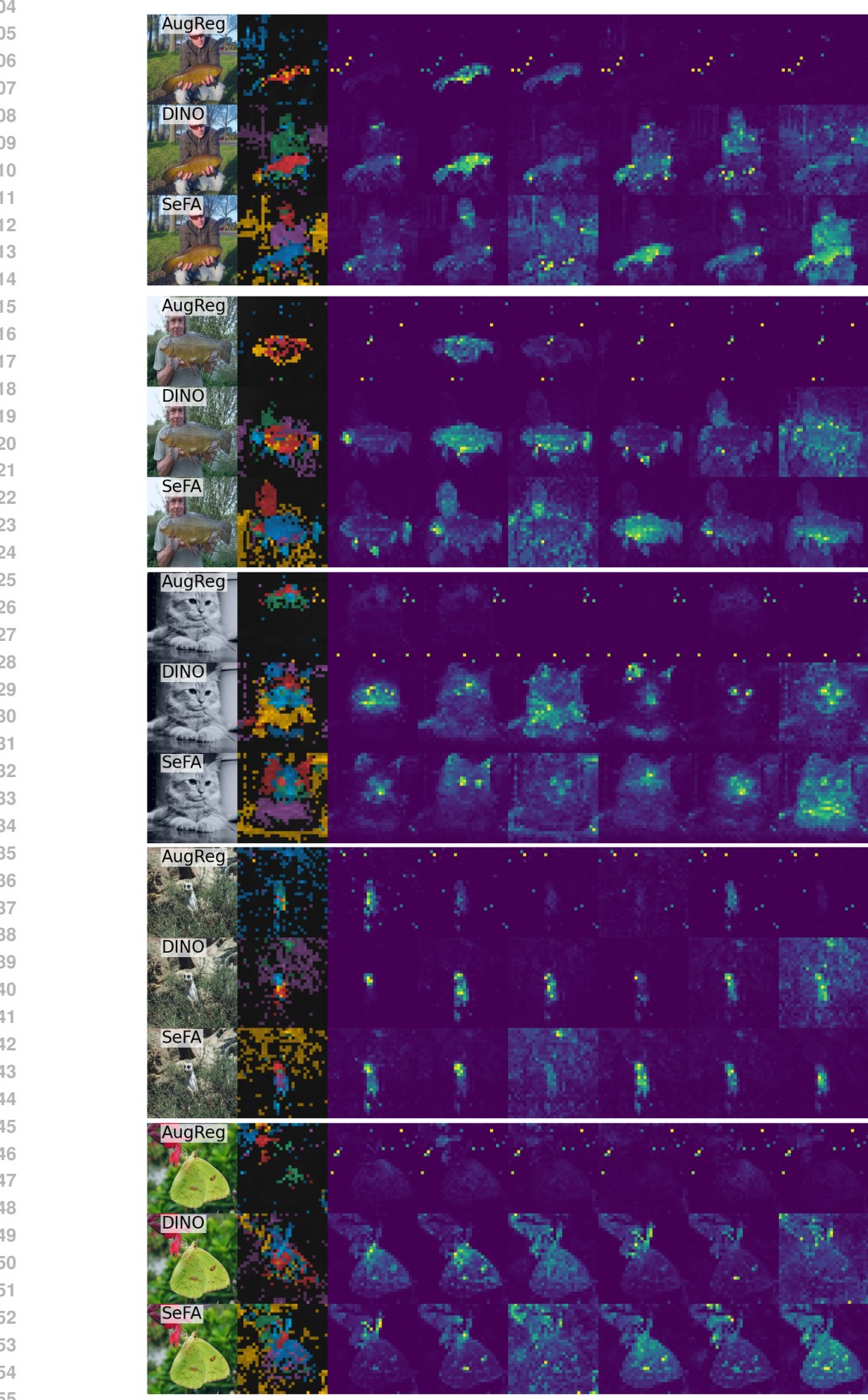

Figure 3: Visualizations of attention maps for SeFA trained with different teachers (AugReg+DINO), (AugReg+MUGS) and (AugReg+iBOT). We provide further visualizations in Table 6.

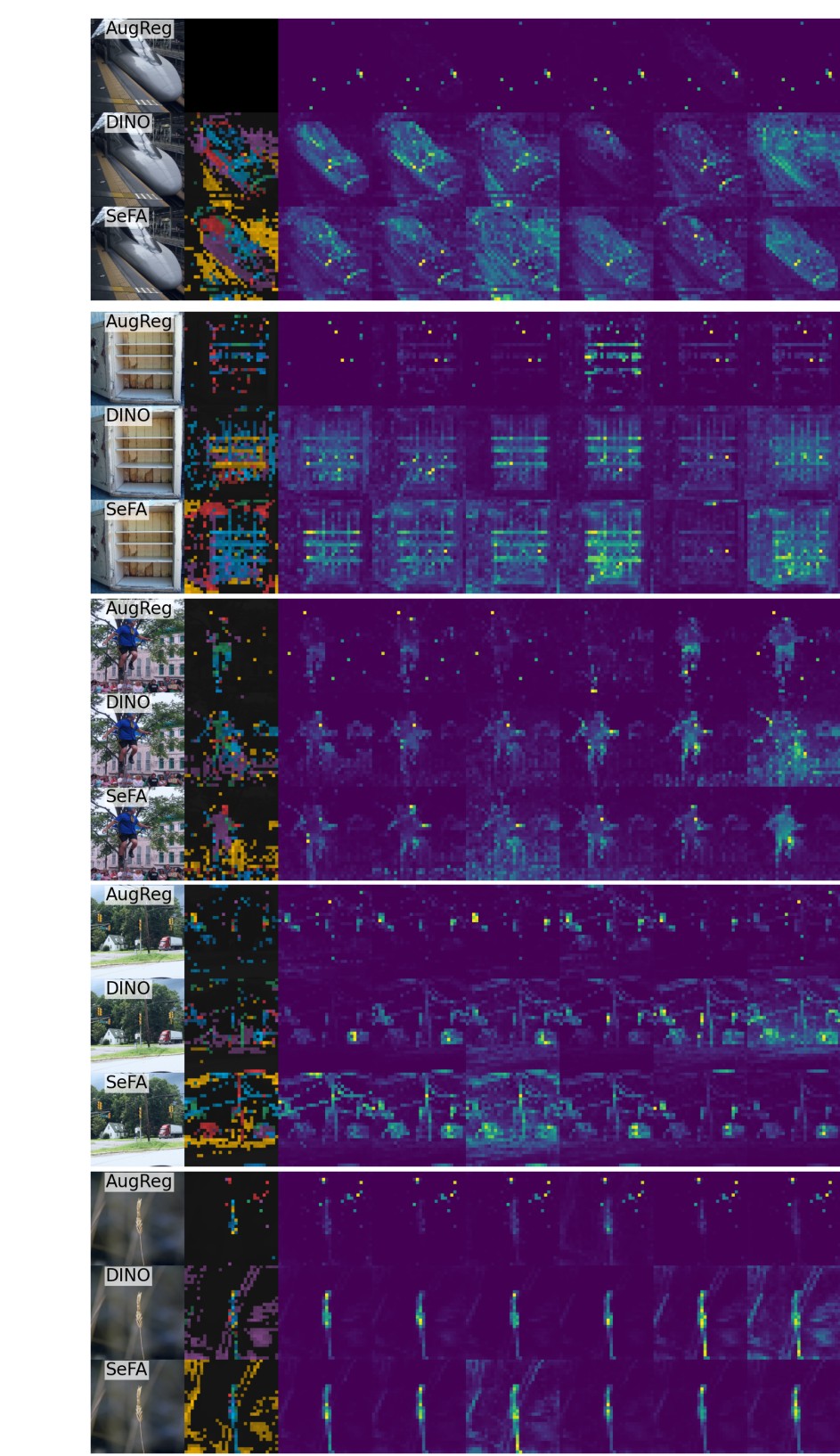

Figure 4: Visualizations of attention maps for SeFA trained with different teachers (AugReg+DINO), (AugReg+MUGS) and (AugReg+iBOT). We provide further visualizations in Table 6.

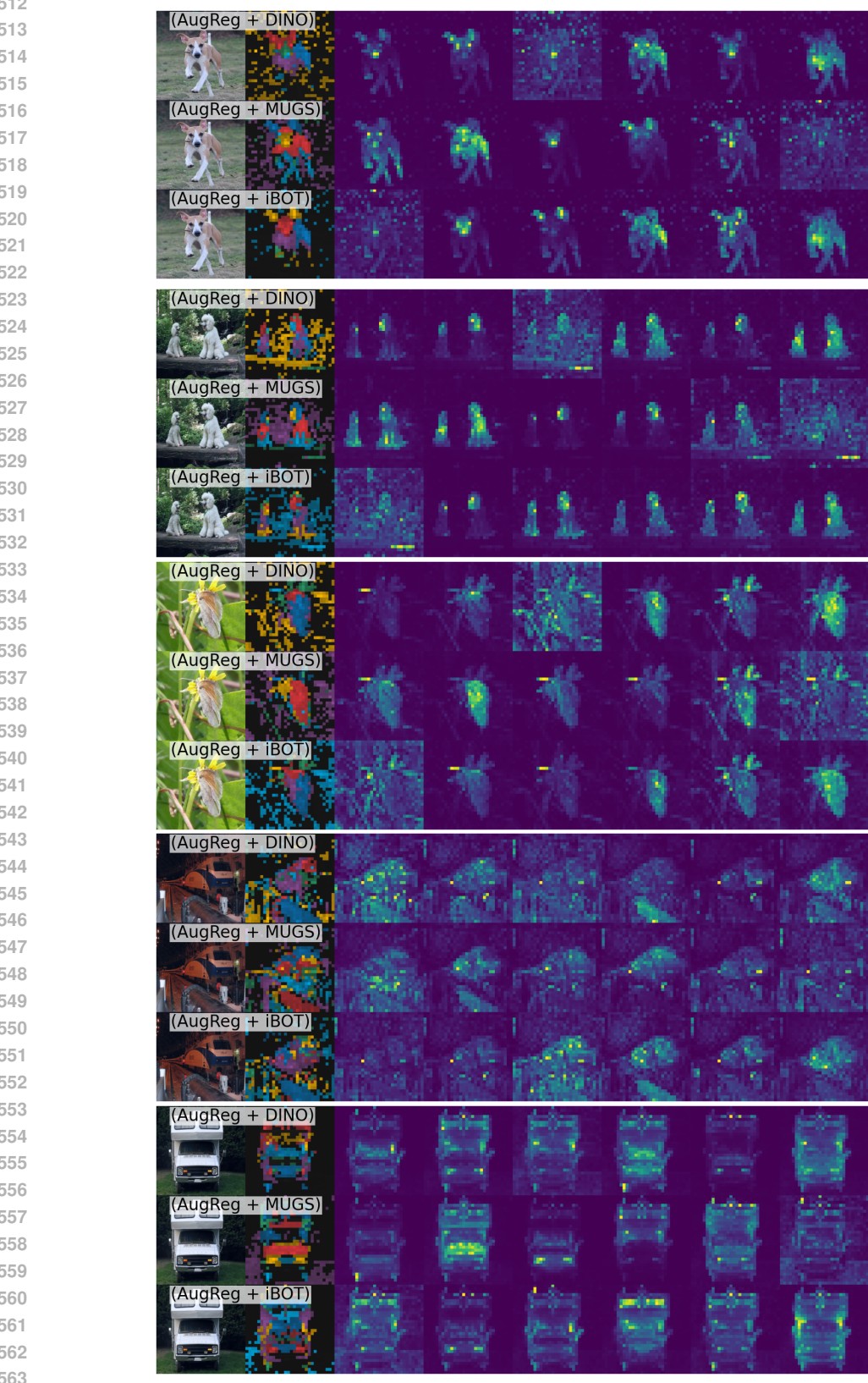

Figure 5: Visualizations of attention maps for SeFA trained with different teachers (AugReg+DINO), (AugReg+MUGS) and (AugReg+iBOT). We provide further visualizations in Table 6.

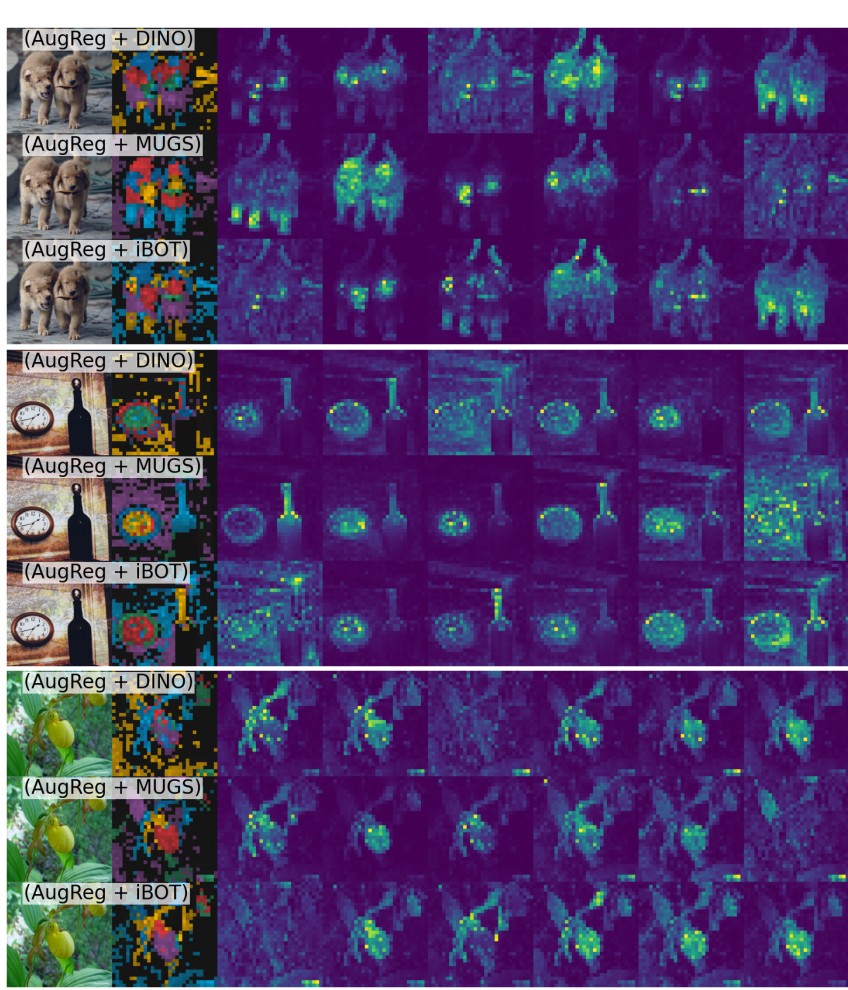

Figure 6: Continued visualizations of attention maps from Table 5 for SeFA trained with different teachers (AugReg+DINO), (AugReg+MUGS) and (AugReg+iBOT).

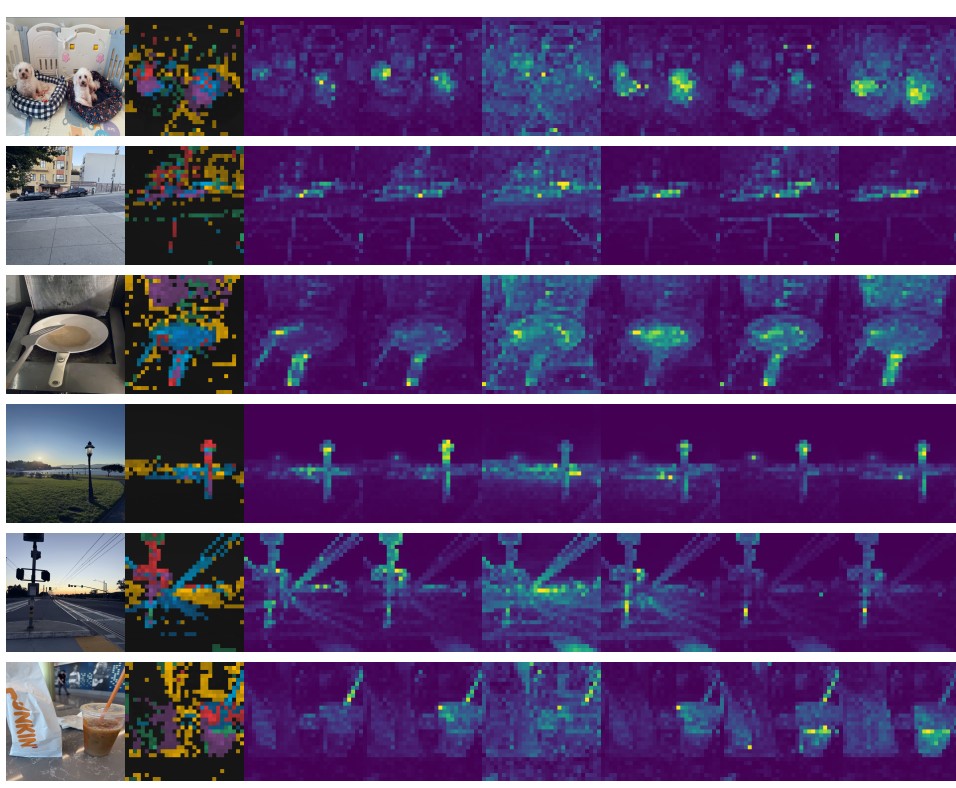

Figure 7: Attention map visualizations for SeFA-trained ViT-S/16 models distilled from ViT-S/16 teachers on real-world images captured with a smartphone camera.

