# OpenReview forum: "Self-Supervision Improves Multi-Teacher Distillation"
_ICLR.cc/2026/Conference — ICLR 2026 Conference Withdrawn Submission_

### Official Review · Reviewer_Ti4m · 2025-10-28

**Soundness:** 2
**Presentation:** 3
**Contribution:** 2
**Rating:** 4
**Confidence:** 4

**Summary:**

The paper proposes Self-supervised Feature Aggregation (SeFA), a novel framework that integrates multi-teacher knowledge distillation with self-supervised learning. Traditional multi-teacher distillation methods constrain the student to learn from teacher representations, limiting the ability to capture useful data-driven features. SeFA overcomes this limitation by framing the task as a multi-task optimization problem, where the student model must learn representations that are both aligned with teacher models and effective for a self-supervised task. Extensive evaluations on various tasks (image classification, transfer learning, image retrieval, dense prediction) show that SeFA outperforms state-of-the-art baselines, achieving improvements in multiple benchmarks.

**Strengths:**

Originality: The combination of multi-teacher distillation with self-supervised learning is a novel approach.

Quality: The paper offers a robust empirical evaluation across multiple tasks, with significant improvements over current state-of-the-art methods.

Clarity: The writing is clear, and the technical details are well-explained, although some parts of the experimental setup could benefit from additional explanation.

Significance: The method has the potential to significantly impact both knowledge distillation and self-supervised learning domains, particularly in vision tasks.

**Weaknesses:**

The method could benefit from a more detailed discussion of its limitations, such as its computational cost or applicability to different model architectures.

Further comparison with other self-supervised approaches would help establish SeFA’s relative strengths.

A clearer theoretical justification for why self-supervision works synergistically with multi-teacher distillation could strengthen the argument.

The formatting is generally good, but the paper could use clearer transitions between the theoretical motivation and the experimental results. Some technical terms could be defined more explicitly for readers unfamiliar with distillation and self-supervised learning techniques.

**Questions:**

1 Could SeFA be applied effectively to domains outside computer vision, such as natural language processing or speech recognition?

2 How do the computational requirements of SeFA compare to traditional distillation methods?

---

### Official Review · Reviewer_KKHc · 2025-10-30

**Soundness:** 3
**Presentation:** 3
**Contribution:** 1
**Rating:** 0
**Confidence:** 5

**Summary:**

This paper introduces Self-supervised Feature Aggregation (SeFA), a framework for improving multi-teacher knowledge distillation by integrating self-supervised learning. The student model simultaneously aligns with teacher representations and learns task-agnostic, data-driven features through a self-supervised task. The proposed loss function integrates a distillation loss with a self-supervised loss, balancing the contributions from both. Experimental results show that the proposed method achieves performance gains over multiple computer vision tasks.

**Strengths:**

1.	The paper is well-organized, with a logical flow from the introduction to the conclusion. Each section builds upon the previous one, making it easy for readers to follow the motivation, methodology, experiments, and results.
2.	The method is simple and easy to understand.
3.	Experimental results show that the proposed method achieves performance gains over multiple computer vision tasks.

**Weaknesses:**

1.	The proposed method is not novel. The proposed method compared to previous UNIC is only adding a self-supervised task loss, which is a general practice, and makes no insights to self-supervised distillation community.
2.	The experimental results are not convincing compared to UNIC paper. UNIC reports 83.2\% accuracy, but the proposed method only shows 80.71\%. This paper should conduct a fair comparison with the UNIC paper.
3.	This paper primarily investigates the integration of self-supervised learning with multi-teacher knowledge distillation. However, there is insufficient comparison with current self-supervised learning with knowledge distillation [1,2,3] and their multi-teacher counterparts.

Reference：

[1] Fang Z, Wang J, Wang L, et al. Seed: Self-supervised distillation for visual representation[J]. arXiv preprint arXiv:2101.04731, 2021.

[2] Bhat P, Arani E, Zonooz B. Distill on the go: Online knowledge distillation in self-supervised learning[C]//Proceedings of the IEEE/CVF Conference on computer vision and pattern recognition. 2021: 2678-2687.

[3] Jang J, Kim S, Yoo K, et al. Self-distilled self-supervised representation learning[C]//Proceedings of the IEEE/CVF winter conference on applications of computer vision. 2023: 2829-2839.

**Questions:**

Please refer to the weakness section.

---

### Official Review · Reviewer_gZ3j · 2025-10-30

**Soundness:** 2
**Presentation:** 2
**Contribution:** 1
**Rating:** 2
**Confidence:** 4

**Summary:**

The paper proposes Self-supervised Feature Aggregation (SeFA), a framework combining self-supervised learning with multi-teacher knowledge distillation. Unlike prior methods such as UNIC, which rely solely on teacher representations, SeFA integrates a self-supervised loss to guide the student toward more generalizable and data-driven representations. The approach achieves consistent performance improvements across image classification, retrieval, and segmentation tasks using ViT-based models

**Strengths:**

The integration of self-supervision into multi-teacher distillation is a novel and well-motivated idea that addresses a clear limitation of existing frameworks.

**Weaknesses:**

The study focuses exclusively on ViT backbones, limiting generality across architectures. The comparison to other multi-teacher or self-distillation baselines is narrow—only UNIC is deeply analyzed. Some sections (e.g., Eq. 2, typographical inconsistencies) indicate incomplete proofreading. Finally, while the contribution is incremental in concept, the novelty lies more in the combination than in a fundamentally new mechanism.

**Questions:**

The introduction does not explicitly list key contributions—clarifying them in a separate paragraph would improve readability.

Why were only ViT-based teachers and students used? Including CNNs could enhance generalizability.

Given that multiple multi-teacher frameworks exist, why was UNIC the sole comparison baseline?

---

### Official Review · Reviewer_HM6c · 2025-11-03

**Soundness:** 3
**Presentation:** 2
**Contribution:** 3
**Rating:** 6
**Confidence:** 2

**Summary:**

The paper introduces SeFA (Self-supervised Feature Aggregation), a new framework that unifies multi-teacher knowledge distillation (KD) with self-supervised learning (SSL).
Traditional multi-teacher distillation constrains a student model to mimic teacher representations, which can limit generalization and cause over-alignment with teacher biases.
SeFA addresses this limitation by reformulating the distillation process as a multi-task optimization problem, combining: (1) a distillation loss  that aligns the student’s features with those of multiple teachers through cosine and Smooth-L1 similarity, and (2) a self-supervised loss (DINO-style cross-view consistency between a student and its EMA copy) that provides data-driven supervision independent of the teachers.
The resulting joint objective encourages the student to learn both teacher-consistent and data-grounded features.
The method requires no additional labels and can be applied to any set of heterogeneous teachers.
Empirically, SeFA is pretrained on ImageNet-1K without labels using ViT-S and ViT-B backbones and evaluated through frozen-encoder linear probing and transfer tasks. The method consistently outperforms the state-of-the-art UNIC baseline and its individual teachers across a wide range of tasks, including classification, transfer learning, image retrieval, and dense prediction (video segmentation).
SeFA achieves up to +1.5 % Top-1 improvement over the best teacher on ImageNet, +6–10 pp gains in robustness benchmarks (ImageNet-R/C), and notable improvements in retrieval and segmentation.
A CKA-based analysis demonstrates that performance correlates with teacher diversity—best results occur when teachers have moderate representational similarity.
Overall, SeFA provides a conceptually simple yet effective way to integrate teacher-driven and data-driven supervision, yielding students that generalize better than their teachers and advancing the field of multi-teacher distillation.

**Strengths:**

1. Conceptual originality.
The paper proposes a simple but conceptually clear extension to multi-teacher distillation: the integration of a self-supervised objective that leverages the intrinsic structure of the unlabeled data. Framing distillation as a multi-task optimization problem, balancing teacher alignment and self-supervised consistency through a single parameter $\lambda_1$, is elegant and intuitively motivated.
Although the idea builds upon known paradigms (multi-teacher KD and DINO-style SSL), its combination is novel in this specific context and provides a clean, reproducible baseline for future hybrid approaches.

2. Strong and diverse empirical evidence.
The experiments are extensive and show consistent improvements over strong baselines such as UNIC and the individual teachers across multiple domains. Gains are reported for ImageNet classification, robustness benchmarks (ImageNet-R/C/A), image retrieval (ROxford, RParis), video segmentation (DAVIS-2017), and transfer learning (CIFAR-10/100, Flowers-102, Pets). The method performs robustly on both ViT-S and ViT-B architectures, confirming scalability across model sizes.

**Weaknesses:**

1. Overstated experimental scope and incomplete coverage of claims.
The paper’s abstract and introduction promise evaluation across “classification, transfer learning, domain adaptation, image retrieval, and dense prediction.” In reality, only four of these categories are presented (classification, transfer, retrieval, and dense prediction) while domain adaptation is not explicitly tested.
Robustness tests on ImageNet-R/C/A partially approximate domain shift but do not constitute a proper adaptation experiment.
Moreover, although the introduction claims to ablate the number and size of teachers, all main-text experiments use a fixed two-teacher configuration (DINO + AugReg). The authors mention in the end that additional experiments with more teachers appear in Appendix C, but such results are central to the paper’s claim of scalability and should have been summarized in the main body.
Relegating them to the appendix undermines the “comprehensive analysis” narrative.

2. Inconsistency between reported best model and analyzed configuration.
Table 1 identifies SeFA + UDI as the top-performing variant, yet all subsequent experiments and analyses are conducted with SeFA + DINO. The authors later justify this by citing computational efficiency (“DINO is the most compute-efficient to train followed by iBOT and UDI”), which is reasonable, but this rationale should appear near Table 1 to preserve continuity and avoid confusion regarding which variant underlies the reported downstream results.

3. Presentation and structural issues.
The paper suffers from inconsistent table ordering and naming conventions (e.g., models appear as “SeFA,” “SeFA (ours),” or “AugReg ViT-S/16”) and some tables are referenced non-sequentially (Table 1 cited after Table 2). Figure 1 includes labeling errors (both branches marked t₂) and does not explicitly depict gradient flow; the “//” stop-gradient symbol should be explained in the caption.
Such inconsistencies make it difficult to track results and give the impression of a rushed submission.

4. Stylistic and editorial shortcomings.
The manuscript contains numerous typographical and grammatical errors (e.g., “section X,” “random hyperparamter,”, “comabinations,” “mentioend”), as well as awkward or incomplete sentences (“With SeFA the loss function to be as follows”).
These do not affect the technical validity but substantially reduce readability and professionalism. Careful proofreading and language editing are required before publication.

5. Minor robustness limitation.
Although SeFA improves generalization across most benchmarks, its performance on adversarially perturbed images (ImageNet-A) remains below that of the best teacher, indicating that robustness gains are not uniform.

**Questions:**

See Weaknesses.

---

### Note · Authors · 2025-11-12

I have read and agree with the venue's withdrawal policy on behalf of myself and my co-authors.